# ASSIGNING CONFIDENCE: K-PARTITION ENSEMBLES

## ABSTRACT

Clustering is widely used for unsupervised structure discovery, but it offers no clear measure of how reliable each individual assignment is. While convergence or objective scores may reflect global quality, they do not indicate whether specific points are stably assigned, particularly in algorithms like $k$-means, which are sensitive to initialization and noise. This assignment-level instability can undermine clustering accuracy and robustness. Ensemble methods improve global consistency by aggregating multiple runs, but they typically lack tools for quantifying pointwise confidence. We introduce **CAKE** (Confidence in Assignments via K-partition Ensembles), a unified framework that evaluates each point using two complementary statistics, assignment stability and consistency of local geometric fit, measured across a clustering ensemble. These are combined into a single interpretable confidence score in $[0, 1]$. Theoretical analysis shows that CAKE scores are robust to noise and reliably distinguish stable from unstable points. Empirical results on synthetic and real-world datasets demonstrate that CAKE identifies both high- and low-confidence assignments, enabling targeted filtering or prioritization that improves clustering quality.

## 1 INTRODUCTION

Clustering is a fundamental task in unsupervised machine learning, widely used to uncover structure in unlabeled data (Jain et al., 1999; Aggarwal & Reddy, 2013). It has core applications in pattern discovery, exploratory analysis, and decision-making across both scientific and applied domains (Kaufman & Rousseeuw, 2009). However, unlike supervised learning, where confidence estimation techniques such as conformal prediction (Shafer & Vovk, 2007) or calibrated probabilities (Guo et al., 2017) are common, clustering methods typically do not offer reliable confidence scores for individual data point assignments. This makes it difficult to assess assignment trustworthiness, which is particularly important when downstream decisions rely on clustering results.

In practice, clustering algorithms often exhibit sensitivity to initialization, susceptibility to local optima, and vulnerability to noisy or ambiguous data (Xu & Wunsch, 2005). Consequently, clustering outcomes may vary across executions, even under fixed algorithmic settings. For example, $k$-means (MacQueen, 1967), despite its popularity, can produce divergent results across runs due to random initialization (Bubeck et al., 2009). Practitioners often mitigate this by running the algorithm multiple times with different random seeds and selecting the best result based on internal validation metrics. However, this approach only addresses global variability and offers no insight into the reliability of individual point assignments. Such assignment-level instability is particularly problematic in settings like anomaly detection or scientific discovery, where unreliable assignments can mislead interpretation or obscure important patterns. Yet, most existing evaluation methods assess clustering quality at a global or cluster level, using internal validation metrics such as the Silhouette score or Davies–Bouldin index (Arbelaitz et al., 2013; Vendramin et al., 2009). Ensemble-based techniques like consensus clustering (Strehl & Ghosh, 2002) have also been explored to improve robustness by aggregating multiple partitions. However, they still lack interpretable, pointwise confidence scores that reflect the trustworthiness of individual assignments. This raises a central question: *How can we assign each data point a confidence value that reflects the quality of its clustering behavior—in terms of both geometric fit and statistical consistency?*

To address this question, we introduce **CAKE** (Confidence in Assignments via K-partition Ensembles), a framework that quantifies per-point confidence by combining two complementary statistics derived from an ensemble of clustering partitions: (*i*) pairwise assignment agreement, computed

via optimal label alignment using the Hungarian algorithm (Meila, 2007; Kuhn, 1955), and (*ii*) local geometric consistency, measured through aggregated Silhouette statistics. These components are integrated into a single interpretable score in $[0, 1]$ that reflects how confidently a point can be assigned to a cluster. Unlike prior ensemble or stability approaches, CAKE yields a numerically interpretable score for every point, enabling fine-grained trust assessment. We provide theoretical guarantees for the statistical reliability of CAKE scores and demonstrate, across synthetic and real-world datasets, that CAKE reliably distinguishes between high- and low-confidence assignments. This enables selective filtering or prioritization of points, improving clustering robustness, interpretability, and quality—all without requiring access to labels. CAKE introduces a principled framework for per-instance unsupervised confidence estimation, bridging a long-standing gap between ensemble diversity and assignment-level trust. By quantifying uncertainty without labels, it advances the foundation for reliable, label-free inference in unsupervised learning.

Our code on CAKE is available at https://anonymous.4open.science/r/cake_confidence-ICLR/.

## 2 RELATED WORK

Assessing the quality of clustering results has been the focus of extensive prior work. Classical validation metrics such as the Silhouette score (Rousseeuw, 1987; Dudek, 2020; Pavlopoulos et al., 2025), Davies–Bouldin index (Davies & Bouldin, 1979), and Calinski–Harabasz criterion (Caliński & Harabasz, 1974) provide global or cluster-level assessments of compactness and separation, but are not suited to identify unstable or ambiguous samples within a partition (Arbelaitz et al., 2013).

Clustering ensembles aim to improve robustness by aggregating multiple partitions, using techniques such as co-association matrices or consensus functions (Strehl & Ghosh, 2002; Fred & Jain, 2005) and more recent ensemble-selection approaches (Golalipour et al., 2021). While effective at stabilizing global structure, these methods typically do not provide per-point confidence scores. Their focus lies in producing a single consensus clustering, not in quantifying how trustworthy each individual assignment is across runs. Moreover, the consensus output may mask uncertainty or variability in specific regions of the data space, making it difficult to identify borderline or unstable points.

Efforts to incorporate uncertainty into clustering include fuzzy clustering (Bezdek, 1981; Bezdek et al., 1984), probabilistic mixture models, and bootstrap-based stability analysis (Hennig, 2007; Lange et al., 2004; Ben-Hur et al., 2002). Recent deep clustering work also studies calibrated cluster confidence, for example via deep clustering networks explicitly trained for confidence calibration (Jia et al., 2025). These approaches provide soft assignments or probability estimates, but often lack interpretability when applied to hard clustering results, and typically do not offer the ensemble-based robustness guarantees; in particular, resampling/label-alignment methods quantify per-point agreement but omit local geometric fit, lacking a unified stability–geometry measure. In addition, model-based techniques such as Gaussian Mixture Models (Reynolds, 2009) require distributional assumptions, while bootstrapping methods can be computationally intensive and sensitive to sampling variability. Another relevant area involves the challenge of label alignment when comparing clustering results, often addressed using the Hungarian algorithm for optimal permutation matching. While this alignment step is standard in ensemble clustering, it has rarely been used to systematically quantify per-point assignment stability across an ensemble.

Extending these approaches, CAKE combines optimal label alignment with local geometric analysis to provide a principled, interpretable confidence score for each data point, bridging ensemble-based robustness with pointwise reliability. CAKE draws inspiration from supervised ensemble techniques, where disagreement between classifiers informs uncertainty (Lakshminarayanan et al., 2017; Gal & Ghahramani, 2016; Beluch et al., 2018). These approaches leverage variation across independently trained models or stochastic passes (such as dropout) to estimate predictive confidence. Unlike these, CAKE operates in an unsupervised regime, using clustering ensemble diversity and within-cluster geometric coherence to infer confidence scores without labels.

## 3 METHODOLOGY

### 3.1 SETTING & NOTATION

Assume a dataset $\mathcal{X} = \{x_i\}_{i=1}^n \subset \mathbb{R}^d$ consisting of $n$ data points. Let $\mathcal{C} = \{C^{(1)}, C^{(2)}, \ldots, C^{(R)}\}$ denote a collection of $R$ clustering results obtained by repeatedly applying the same clustering algorithm to $\mathcal{X}$, each time using the same number of clusters $k$ but with a different random seed.

This collection forms a clustering ensemble by capturing multiple partitions of the dataset under stochastic variation. Although all runs use the same algorithm and parameters, random initialization introduces meaningful differences in the resulting partitions, reflecting uncertainty in how individual points are grouped. More generally, the ensemble could be constructed using resampling strategies (e.g., clustering on bootstrapped subsets of $\mathcal{X}$), or by aggregating partitions from different clustering algorithms, offering a broader perspective on assignment variability. While our framework is compatible with such (heterogeneous) ensemble construction strategies, including resampling and multi-algorithm clustering ensembles, in this work, we focus on (homogeneous) ensembles formed by repeated applications of a single clustering method, allowing us to quantify assignment-level confidence without interference from other sources of variation. To work with these partitions quantitatively, we define $\mathcal{L} = \left\{ L^{(1)}, L^{(2)}, \ldots, L^{(R)} \right\}$ as the corresponding set of label assignments. Each labeling $L^{(r)} \in \mathcal{L}$ assigns a cluster label to every data point $x_i \in \mathcal{X}$, such that: $L^{(r)} = \left\{ L_1^{(r)}, L_2^{(r)}, \ldots, L_n^{(r)} \right\}$, where $L_i^{(r)} \in \{1, 2, \ldots, k\}$ denotes the label of $x_i$ in clustering $r$.

## 3.2 Ensemble Silhouette Statistics

For each point $x_i \in \mathcal{X}$, let $s_i^{(r)}$ denote its Silhouette score under clustering run $r \in \{1, 2, \ldots, R\}$, $s_i^{(r)} = \left( b_i^{(r)} - a_i^{(r)} \right) / \max \left\{ a_i^{(r)}, \, b_i^{(r)} \right\} \in [-1, 1]$, which quantifies the quality of its assignment by comparing its average intra-cluster distance $a_i^{(r)}$ to its minimum average inter-cluster distance $b_i^{(r)}$. Computing all $s_i^{(r)}$ is $O(n^2 d)$ per clustering run. When the base method is centroidal (e.g., $k$-means / $k$-means++) and $n$ is large, we compute a centroid-based Silhouette approximation (Wang et al., 2017) by replacing $\tilde{a}_i^{(r)} = \|x_i - \mu_{C_i}^{(r)}\|$ and $\tilde{b}_i^{(r)} = \min_{C \neq C_i} \|x_i - \mu_C^{(r)}\|$, where $\mu_C^{(r)}$ is the centroid of cluster $C$ in run $r$, and $C_i$ denotes the cluster of $x_i$ in run $r$. This reduces the per–run cost to $O(nkd)$ while tracking the exact score closely in centroidal settings.

**Aggregating over the ensemble.** We compute the mean Silhouette score $\mu_i = \frac{1}{R} \sum_{r=1}^{R} s_i^{(r)}$ and standard deviation (std) $\sigma_i = \sqrt{\frac{1}{R} \sum_{r=1}^{R} \left( s_i^{(r)} - \mu_i \right)^2}$ for each point $x_i$ across all $R$ clustering runs in the ensemble. We use these two quantities to define a per-point measure of Silhouette-based reliability. A high mean $\mu_i$ indicates that $x_i$ tends to be well-placed geometrically across clustering runs, while a low std $\sigma_i$ implies that this quality is consistent. Subtracting $\sigma_i$ from $\mu_i$ penalizes points with high variability in Silhouette values across runs, emphasizing both quality and stability of geometric fit. The resulting score is thresholded at zero to ensure non-negativity[1]:

$$\tilde{S}_i = (\mu_i - \sigma_i)_+ = \max \left\{ 0, \, (\mu_i - \sigma_i) \right\} \in [0, 1]. \tag{1}$$

$\tilde{S}_i$ rewards points that not only have high average Silhouette (good cluster fit) but also low variability across runs (stable geometric placement). However, a high $\tilde{S}_i$ does not guarantee that a point is consistently assigned to the same cluster across runs. It only indicates that the point maintains a consistently strong fit within its assigned cluster. To capture actual assignment stability, we introduce a measure based on cluster agreement across the ensemble.

## 3.3 Ensemble Assignment Stability

**Label alignment.** Given two clustering labelings $L^{(r_1)}, L^{(r_2)} \in \mathcal{L}$ from clustering runs $r_1$, $r_2$, the labels may not be aligned: that is, cluster label $j$ in $L^{(r_1)}$ and in $L^{(r_2)}$ may correspond to a different group of points. To align $L^{(r_2)}$ onto $L^{(r_1)}$, we define a contingency matrix $M \in \mathbb{N}^{k \times k}$, where each entry $M_{i,j}$ counts the number of points assigned to cluster $i$ in $L^{(r_1)}$ and cluster $j$ in $L^{(r_2)}$:

$$M_{i,j} = \sum_{p=1}^{n} \mathbb{1} \left\{ \left( L_p^{(r_1)} = i \right) \wedge \left( L_p^{(r_2)} = j \right) \right\} \tag{2}$$

where $\mathbb{1}(\cdot)$ is the indicator function. Finding the optimal alignment corresponds to finding a permutation $P^* : \{1, \ldots, k\} \to \{1, \ldots, k\}$ that maximizes agreement: $P^* = \arg\max_P \sum_{i=1}^{k} M_{i, P(i)}$,

---

[1]To preserve the signal from consistently negative means (e.g., $\mu_i < 0$ with small $\sigma_i$; a relatively uncommon regime in practice), $\mu_i - \sigma_i$ can be mapped to $[0, 1]$ via $((\mu_i - \sigma_i) + 1)/2$.

where $P(i)$ denotes the label in $L^{(r_2)}$ that is matched to label $i$ in $L^{(r_1)}$. This optimization is equivalent to solving a linear sum assignment problem, which can be solved using the Hungarian algorithm in $O(k^3)$ time. After aligning $L^{(r_2)}$ onto $L^{(r_1)}$, we denote the aligned labeling by $L^{(r_2 \to r_1)}$ and define the pointwise agreement between two clustering runs $(r_1, r_2)$ for a data point $x_i \in \mathcal{X}$ as:

$$A_i^{(r_1, r_2)} = \mathbb{1}\left\{ L_i^{(r_1)} = L_i^{(r_2 \to r_1)} \right\} \in \{0, 1\} \tag{3}$$

**Instance pairwise stability score.** Given $R$ clustering runs, there are $\binom{R}{2} = \frac{R(R-1)}{2}$ distinct unordered pairs of partitions. We define the pairwise stability score $c_i$ for every point $x_i \in \mathcal{X}$ as the fraction of all unordered run-pairs in which $x_i$ is assigned to the same cluster (after alignment):

$$c_i = \frac{2}{R(R-1)} \sum_{r_1 < r_2} A_i^{(r_1, r_2)} \tag{4}$$

By definition, $c_i$ ($\in [0, 1]$) rewards points that are consistently assigned to the same cluster across the ensemble. Points with high $c_i$ exhibit minimal assignment variation and can be viewed as stable members of the underlying clustering structure. Moreover, since $A_i^{(r_1, r_2)}$ is a Bernoulli indicator taking values in $\{0, 1\}$, the quantity $c_i$ is an order-2 U-statistic with a binary (Bernoulli) kernel (Lee, 1990). As such, $c_i$ is an unbiased estimator of the true pairwise assignment stability $\mathbb{E}[A_i^{(r_1, r_2)}]$.[2]

## 3.4 CAKE SCORE

From the clustering ensemble, we derive two complementary per-instance diagnostics: the assignment-stability estimate $c_i$ (Eq. 4) and the Silhouette-based reliability score $\tilde{S}_i$ (Eq. 1). The CAKE score combines these signals into a single confidence value in $[0, 1]$ (outlined as pseudocode in Appendix A). It rewards observations that are repeatedly assigned to the same cluster across runs and are strongly supported by their local geometry, while penalizing points with unstable assignments or poor local separation from other clusters. Our two formulations are:

$$\text{CAKE}_i^{(\text{PR})} = c_i \, \tilde{S}_i \quad (5a) \qquad\qquad \text{CAKE}_i^{(\text{HM})} = \frac{2 \, c_i \, \tilde{S}_i}{c_i + \tilde{S}_i} \quad (5b)$$

The product (Eq. 5a) attains high values only when both components are large and drops quickly if either is weak; the harmonic mean (Eq. 5b) is less punitive when one term is moderate, which can aid interpretability and calibration. Notably, this mirrors the F1 score in classification, which harmonizes precision and recall (Powers, 2011), reinforcing CAKE's interpretability as a principled confidence measure (see also spectral clustering ensemble approach for non-convex data in Appendix C.4).

**Complexity.** With the centroid proxy, computing Silhouette scores over $R$ runs costs $O(Rnkd)$, while exact computations cost $O(Rn^2 d)$. Label alignment over all run-pairs builds a $k \times k$ contingency and solves a Hungarian assignment per pair, for $O\left(\binom{R}{2}(n + k^3)\right)$ time; thus CAKE runs in $O\left(Rnkd + \binom{R}{2}(n + k^3)\right)$ overall, or $O\left(Rn^2 d + \binom{R}{2}(n + k^3)\right)$ using exact Silhouette scores, and uses $O(nR)$ memory (see runtime vs. $R, n$ on synthetic data in Fig. 6 and Appendix Table 3).

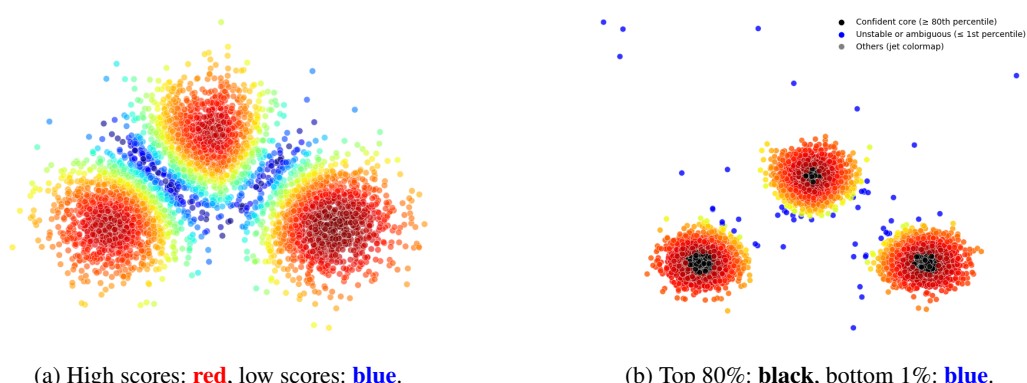

(a) High scores: **red**, low scores: **blue**.      (b) Top 80%: **black**, bottom 1%: **blue**.

Figure 1: CAKE (Eq. 5a) scores distribution (left) and percentiles (right) on synthetic Gaussian data.

---

[2]Expected agreement over pairs of runs, which reflects how reproducibly $x_i$ is clustered across the ensemble.

## 4 THEORETICAL GUARANTEES

We provide two non-asymptotic results for the finite-ensemble stability scores $c_i$; full derivations and analogous guarantees for the geometric component $\tilde{S}_i$ are given in Appendix B.

### 4.1 RANKING-ERROR BOUND FOR FINITE ENSEMBLES

Let $R \geq 2$ be the number of independent clustering runs performed on $\mathcal{X}$. For a point $x_i \in \mathcal{X}$ the true pairwise stability is $\theta_i = \mathbb{E}\left[A_i^{(r_1,r_2)}\right]$, where $A_i^{(r_1,r_2)}$ (Eq. 3) is the Bernoulli kernel used in our order-2 U-statistic estimator $c_i$ (Eq. 4). Intuitively, our empirical scores should rank points in the same order as their true stability $\theta_i$ and any reversed ordering should be random noise that fades when more runs are added.[3] The following bound shows how the misordering probability shrinks with $R$ (derivations in Appendix B.1). For any two points $x_i, x_j \in \mathcal{X}$ and margin $\gamma > 0$, it holds:

$$\mathbb{E}\left[A_i^{(r_1,r_2)}\right] \geq \mathbb{E}\left[A_j^{(r_1,r_2)}\right] + \gamma \Rightarrow \Pr[c_i < c_j] \leq 2\exp\left\{-R\frac{\gamma^2}{8}\right\}. \tag{6}$$

### 4.2 FALSE-POSITIVE BOUND FOR UNIFORM-NOISE POINTS

Consider a uniform-noise observation, i.e. a point whose cluster label in each run is drawn independently and uniformly from $\{1, \ldots, k\}$. Then $\theta_i = \frac{1}{k}$ and deviations of $c_i$ from $\theta_i$ decay exponentially in $R$ (Hoeffding, 1963). Thus, the probability that a noise point falsely attains a high score decreases rapidly with ensemble size (detailed analysis in Appendix B.2):

$$\Pr[c_i > \tau \mid x_i \text{ uniform-noise}] \leq \exp\left\{-\frac{R}{2}\left(\tau - \frac{1}{k}\right)^2\right\} \quad \forall \text{ threshold } \tau > \frac{1}{k}. \tag{7}$$

Additionally, if a $\phi$ fraction of the $n$ observations are uniform-noise, the expected count exceeding $\tau > 1/k$ is at most: $n \cdot \phi \cdot \exp\left\{-\frac{R}{2}\left(\tau - \frac{1}{k}\right)^2\right\}$. More generally, for any (possibly non-uniform) label distribution with expected agreement $\theta_i = \mathbb{E}[A_i^{(r_1,r_2)}]$: $\Pr[c_i > \tau] \leq \exp\left\{-\frac{R}{2}(\tau - \theta_i)^2\right\}$, (for any threshold $\tau > \theta_i$) so the probability still decays exponentially in $R$ (Appendix B.2).

## 5 EMPIRICAL VALIDATION

### 5.1 SYNTHETIC & REAL-WORLD DATASETS

We evaluate CAKE on synthetic and real-world datasets.[4] The **synthetic datasets** are described next (visualizations in Appendix C.2; Fig.7). **S1** has 4,000 points sampled from three Gaussian clusters with equal size and std of 2.0; **S2** with 3,000 points from three clusters with std of 2.0, 2.5, and 1.5. **S3** has 4,500 points, where 3,000 belong to three Gaussian clusters with unit std, and 1,500 are uniformly distributed noise points. **S4** has 3,000 points arranged in four wide Gaussian clusters, located at the corners of a square. **S5** has 4,000 points from three clusters with strong density contrast: one tight cluster with std of 0.2, one wide with std of 3.0, and one intermediate with unit std. **S6** has 4,000 points from three clusters with std of 0.4, 2.5, and 0.4, where a sparse central cluster overlaps denser ones on each side. **S7** has 4,000 points, distributed across three clusters of size 500, 1,000, and 2,500, with increasing std of 0.3, 1.5, and 2.5 respectively. The **real-world datasets** span a wide range of domains and structures (sizes, dimensionality, and preprocessing are summarized in Appendix C.1; Table 2): **Iris** (IR), **Breast Cancer** (BC), **Pendigits** (PD), **Letter** (LT), **Digits** (DG), **Fashion MNIST** (FM), **Satimage** (SA), and **20 Newsgroups** (NG).

### 5.2 EVALUATION SETUP

**Instance Removal.** We remove (low-confidence) instances based on their CAKE scores (§3.4), assessing the impact of this removal in clustering quality. For each dataset, we compute CAKE scores

---

[3] By standard concentration inequalities for bounded U-statistics, $c_i \to \theta_i$ exponentially fast with $R$.

[4] All real-world datasets are publicly available via OpenML (Vanschoren et al., 2013), SCIKIT-LEARN (Pedregosa et al., 2011), or TensorFlow datasets (Abadi et al., 2015).

using an ensemble of 20 independent $k$-means runs at the ground-truth number of clusters $k$ (random initialization for synthetic data; $k$-means++ for real data). We use $k$-means because it is widely used, easy to interpret, and known to be sensitive to initialization, making it a natural candidate for assessing assignment-level confidence; CAKE itself is model-agnostic and applies to any hard-assignment clustering ensemble (illustrated on BC and S5 via CAKE score distributions for homogeneous $k$-means / GMM, and heterogeneous $k$-means+GMM ensembles in Appendix C.5, Fig. 12, and on non-convex data via a spectral-clustering ensemble in Appendix C.4, Fig. 8). For each dataset, we form six equal-size subsets, each retaining exactly 70% of the points, $m = \lfloor 0.7n \rfloor$, according to different criteria: **Random** (sample $m$ points uniformly), **Consensus** (top $m$ by across-run label agreement after aligning all runs to a reference–medoid labeling via the Hungarian algorithm), $\tilde{\mathbf{S}}$ (top $m$ by the Silhouette component; Eq. 1), $\mathbf{C}$ (top $m$ by the stability component; Eq. 4), **CAKE$^{(PR)}$** (top $m$ by the product; Eq. 5a), and **CAKE$^{(HM)}$** (top $m$ by the harmonic mean; Eq. 5b). While this specific threshold is not necessarily optimal, it offers a reasonable coverage–confidence trade-off. We compare clustering performance on the full datasets and on the filtered subsets using new $k$-means runs (separate from the CAKE ensemble) at the ground-truth $k$. We measure performance using Adjusted Rand Index (ARI), Adjusted Mutual Information (AMI), and clustering accuracy (ACC; after Hungarian alignment to ground-truth labels). To assess variability, we repeat the evaluation with multiple independent runs and report means with (Student's $t$) 95% confidence intervals (see Table 1). Additional results for Silhouette, Normalized Mutual Information (NMI), and correlation between CAKE scores and clustering accuracy percentiles, together with a comparison to bootstrap-based stability in terms of accuracy correlation, are provided in the Appendix C.5; Table 4, Fig. 13.

**Correlation with Accuracy.** We examine how CAKE scores correlate with instance-level clustering accuracy. Using the previously computed scores, we re-cluster the full datasets to obtain predicted labels, partition points into CAKE deciles, and compute accuracy within each decile by aligning predictions to ground truth, revealing accuracy–CAKE trends (Fig. 2; see also Appendix C.5 Fig. 11 for a coverage–accuracy trade-off comparison of CAKE$^{(PR)}$ and CAKE$^{(HM)}$ on real datasets).

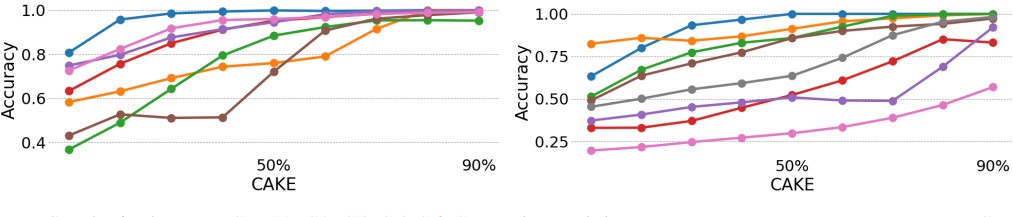

(a) Synthetic datasets: S1, S2, S3, S4, S5, S6, S7.    (b) Real datasets: IR, BC, DG, NG, FM, PD, LT, SA.

Figure 2: Clustering accuracy vs. CAKE$^{(PR)}$ score (Eq. 5a) percentile on synthetic and real datasets.

**Convergence.** We study how CAKE scores stabilize with ensemble size. For each $R \in \{5, \ldots, 70\}$ we draw $B=10$ sub-ensembles of size $R$, recompute CAKE scores using the centroid-based Silhouette proxy (§3.2), and report the per-point standard deviation across the $B$ estimates (Fig. 3).

**Misspecified k.** We test CAKE when the clustering uses a suboptimal number of clusters $k'$. For each dataset and $k' \in \{k-2, \ldots, k+2\}$, where $k$ is the ground-truth number of clusters, we (i) build an ensemble of $R=30$ $k$-means runs at $k'$ and compute CAKE$^{(HM)}$ scores, and (ii) fit a separate $k$-means++ model at $k'$ for evaluation. We report ARI to ground truth and the AURC of CAKE (Fig. 4 and in Appendix C.5 Fig. 9– 10).[5]

**Error discovery via confidence ranking.** We assess CAKE as a confidence score for flagging clustering errors: for each dataset we fit a reference $k$-means++ model at the ground-truth $k$, align its labels to the ground truth via the Hungarian algorithm, and treat misclustered points as positives ("errors"). We then compute three signals, *CAKE$^{(HM)}$* (Eq. 5b; $R=20$ random $k$-means runs with centroid-based Silhouette approximation), *GMM* $p_{max}$ (maximum posterior under a $k$-component Gaussian mixture), and a rank-average *Fusion* of CAKE and $p_{max}$, and rank points from low to high confidence, measuring how well each concentrates errors using AUPRC on the error class (Fig. 5).[6]

---

[5] AURC: area under risk–coverage for CAKE-ranked points based on Hungarian label-aligned correctness.

[6] AUPRC: area under precision–recall for misclustered points ranked by confidence (CAKE scores or $p_{max}$).

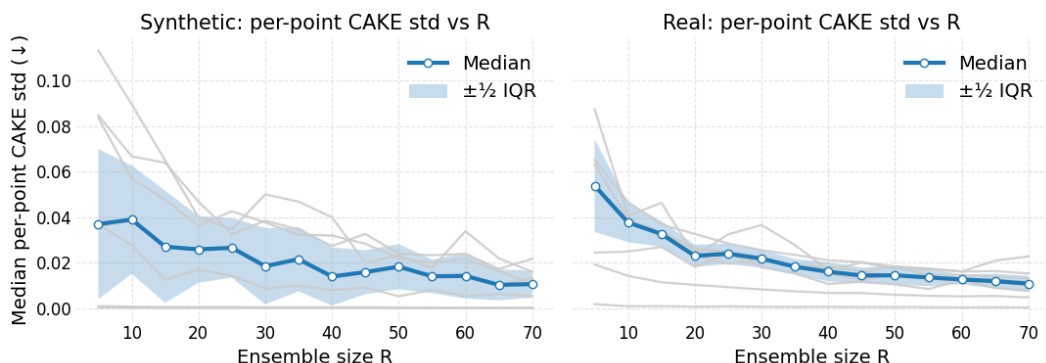

Figure 3: Convergence vs $R$. Median per-point $\text{CAKE}^{(\text{HM})}$ scores (Eq. 5b) std vs. ensemble size $R$ for synthetic (left) and real (right) datasets. **Lines:** light gray: individual datasets; blue: across-dataset median. **Band:** $\pm\frac{1}{2}$ IQR. Variability drops and stabilizes by $R\approx30\text{--}40$ across datasets.

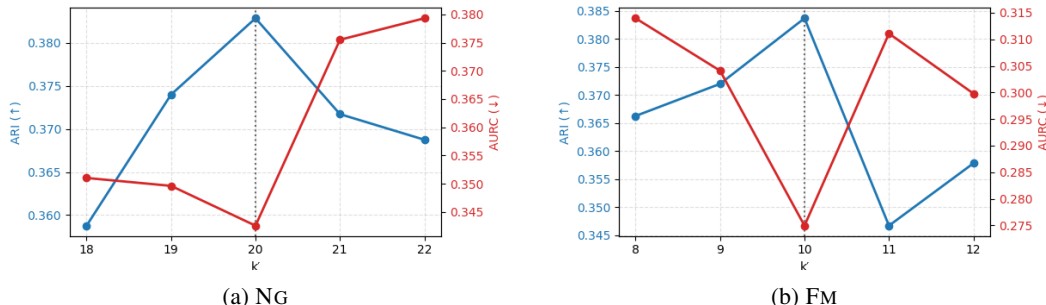

(a) NG                 (b) FM

Figure 4: Misspecified $k$. ARI (blue), AURC (red) vs. $k'$. Vertical dotted line marks the ground-truth $k$. Ensembles use $R=30$ runs to compute $\text{CAKE}^{(\text{HM})}$ scores (Eq. 5b); evaluation uses a separate $k$-means++ at the same $k'$. Lower AURC indicates better CAKE ranking of correct assignments.

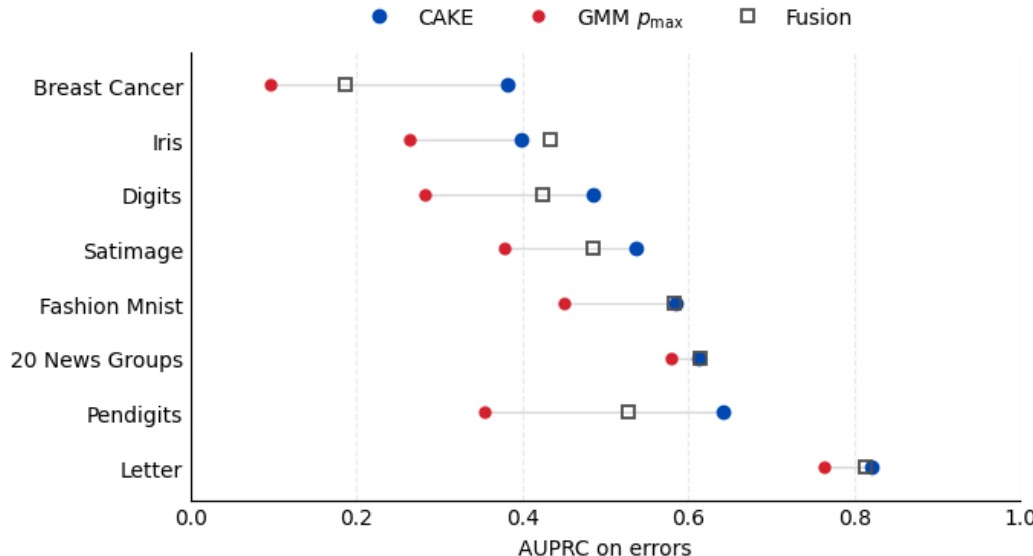

Figure 5: Error discovery via confidence ranking (real datasets) - AUPRC on the error class. Markers show per-dataset AUPRC; light gray connectors join GMM $p_{\max}$ (red) to CAKE (blue) to highlight the per-dataset CAKE score improvement. Across datasets, CAKE concentrates errors more effectively than GMM $p_{\max}$, and the simple fusion (□) is competitive.

| Dataset | Subset | avg ARI | avg AMI | avg ACC |
|---|---|---|---|---|
| S1 | Full | 0.879 [0.8785, 0.8791] | 0.833 [0.8331, 0.8338] | 0.958 [0.9578, 0.9580] |
| | Random | 0.879 [0.8790, 0.8790] | 0.834 [0.8339, 0.8339] | 0.958 [0.9579, 0.9579] |
| | Consensus | 0.884 [0.8842, 0.8842] | 0.839 [0.8394, 0.8394] | 0.960 [0.9596, 0.9596] |
| | $C$ component | 0.884 [0.8842, 0.8842] | 0.839 [0.8394, 0.8394] | 0.960 [0.9596, 0.9596] |
| | $\tilde{S}$ component | **0.992** [0.9921, 0.9921] | **0.983** [0.9827, 0.9827] | **0.997** [0.9971, 0.9971] |
| | CAKE[PR] | **0.992** [0.9921, 0.9921] | **0.983** [0.9827, 0.9827] | **0.997** [0.9971, 0.9971] |
| | CAKE[HM] | **0.992** [0.9921, 0.9921] | **0.983** [0.9827, 0.9827] | **0.997** [0.9971, 0.9971] |
| S2 | Full | 0.552 [0.5225, 0.5812] | 0.606 [0.6056, 0.6068] | 0.762 [0.6993, 0.8244] |
| | Random | 0.529 [0.4901, 0.5684] | 0.601 [0.5923, 0.6102] | 0.724 [0.6441, 0.8029] |
| | Consensus | 0.528 [0.5070, 0.5492] | 0.628 [0.6031, 0.6530] | **0.797** [0.7838, 0.8094] |
| | $C$ component | 0.528 [0.5070, 0.5492] | 0.628 [0.6031, 0.6530] | **0.797** [0.7838, 0.8094] |
| | $\tilde{S}$ component | 0.679 [0.5731, 0.7840] | 0.698 [0.6789, 0.7162] | 0.759 [0.6359, 0.8814] |
| | CAKE[PR] | **0.680** [0.5741, 0.7860] | **0.700** [0.6800, 0.7192] | 0.761 [0.6393, 0.8827] |
| | CAKE[HM] | **0.680** [0.5738, 0.7858] | 0.699 [0.6796, 0.7182] | 0.760 [0.6367, 0.8829] |
| S3 | Full | 0.498 [0.4941, 0.5016] | 0.614 [0.6098, 0.6178] | 0.733 [0.7268, 0.7384] |
| | Random | 0.449 [0.3792, 0.5188] | 0.580 [0.5352, 0.6255] | 0.678 [0.6076, 0.7479] |
| | Consensus | 0.753 [0.7090, 0.7973] | 0.757 [0.7301, 0.7836] | 0.842 [0.8013, 0.8828] |
| | $C$ component | 0.719 [0.6957, 0.7432] | 0.721 [0.7041, 0.7379] | 0.830 [0.8128, 0.8473] |
| | $\tilde{S}$ component | **0.795** [0.7667, 0.8241] | **0.781** [0.7591, 0.8022] | **0.862** [0.8348, 0.8886] |
| | CAKE[PR] | 0.734 [0.6444, 0.8232] | 0.751 [0.6982, 0.8031] | 0.784 [0.7085, 0.8591] |
| | CAKE[HM] | 0.784 [0.7538, 0.8137] | 0.776 [0.7597, 0.7932] | 0.837 [0.8022, 0.8724] |
| S4 | Full | 0.716 [0.7158, 0.7167] | 0.678 [0.6775, 0.6786] | 0.884 [0.8833, 0.8837] |
| | Random | 0.727 [0.7268, 0.7276] | 0.693 [0.6926, 0.6933] | 0.888 [0.8883, 0.8886] |
| | Consensus | 0.606 [0.5946, 0.6179] | 0.598 [0.5939, 0.6029] | 0.755 [0.7432, 0.7665] |
| | $C$ component | 0.606 [0.5946, 0.6179] | 0.598 [0.5939, 0.6029] | 0.755 [0.7432, 0.7665] |
| | $\tilde{S}$ component | **0.902** [0.9022, 0.9022] | **0.867** [0.8667, 0.8667] | **0.962** [0.9624, 0.9624] |
| | CAKE[PR] | **0.902** [0.9022, 0.9022] | **0.867** [0.8667, 0.8667] | **0.962** [0.9624, 0.9624] |
| | CAKE[HM] | **0.902** [0.9022, 0.9022] | **0.867** [0.8667, 0.8667] | **0.962** [0.9624, 0.9624] |
| S5 | Full | 0.593 [0.4434, 0.7435] | 0.660 [0.5739, 0.7462] | 0.782 [0.6384, 0.9254] |
| | Random | 0.549 [0.3891, 0.7089] | 0.634 [0.5415, 0.7267] | 0.743 [0.5933, 0.8933] |
| | Consensus | 0.479 [0.4780, 0.4807] | 0.570 [0.5689, 0.5710] | 0.708 [0.7046, 0.7116] |
| | $C$ component | 0.688 [0.5929, 0.7831] | 0.697 [0.6399, 0.7540] | 0.867 [0.7988, 0.9355] |
| | $\tilde{S}$ component | 0.840 [0.6821, 0.9980] | 0.855 [0.7557, 0.9535] | 0.894 [0.7637, 1.0000] |
| | CAKE[PR] | 0.908 [0.9075, 0.9075] | 0.879 [0.8792, 0.8792] | 0.966 [0.9664, 0.9664] |
| | CAKE[HM] | **0.925** [0.9245, 0.9245] | **0.898** [0.8979, 0.8979] | **0.973** [0.9729, 0.9729] |
| S6 | Full | 0.303 [0.3006, 0.3045] | 0.473 [0.4720, 0.4748] | 0.681 [0.6794, 0.6834] |
| | Random | 0.315 [0.2978, 0.3325] | 0.480 [0.4690, 0.4920] | 0.689 [0.6748, 0.7032] |
| | Consensus | 0.561 [0.5537, 0.5684] | 0.586 [0.5780, 0.5943] | 0.779 [0.7712, 0.7859] |
| | $C$ component | 0.565 [0.5640, 0.5654] | 0.590 [0.5896, 0.5907] | 0.782 [0.7815, 0.7828] |
| | $\tilde{S}$ component | 0.566 [0.5506, 0.5811] | 0.590 [0.5770, 0.6036] | 0.781 [0.7654, 0.7960] |
| | CAKE[PR] | 0.558 [0.5302, 0.5860] | 0.583 [0.5573, 0.6095] | 0.777 [0.7524, 0.8006] |
| | CAKE[HM] | **0.573** [0.5662, 0.5798] | **0.596** [0.5893, 0.6034] | **0.789** [0.7817, 0.7962] |
| S7 | Full | 0.615 [0.4786, 0.7523] | 0.649 [0.5642, 0.7333] | 0.802 [0.6727, 0.9317] |
| | Random | 0.602 [0.4685, 0.7363] | 0.636 [0.5464, 0.7253] | 0.797 [0.6691, 0.9257] |
| | Consensus | 0.460 [0.4371, 0.4820] | 0.545 [0.5385, 0.5519] | 0.679 [0.6411, 0.7165] |
| | $C$ component | 0.450 [0.4267, 0.4725] | 0.540 [0.5326, 0.5480] | 0.668 [0.6329, 0.7035] |
| | $\tilde{S}$ component | 0.713 [0.5401, 0.8869] | 0.762 [0.6529, 0.8713] | 0.820 [0.6817, 0.9582] |
| | CAKE[PR] | **0.861** [0.7566, 0.9662] | **0.861** [0.7908, 0.9309] | **0.939** [0.8647, 1.0000] |
| | CAKE[HM] | 0.678 [0.4982, 0.8581] | 0.741 [0.6239, 0.8579] | 0.797 [0.6623, 0.9308] |
| IR | Full | 0.692 [0.6255, 0.7580] | 0.728 [0.6907, 0.7651] | 0.857 [0.7859, 0.9288] |
| | Random | 0.643 [0.5586, 0.7277] | 0.697 [0.6525, 0.7425] | 0.823 [0.7323, 0.9134] |
| | Consensus | 0.658 [0.5951, 0.7206] | 0.733 [0.7047, 0.7620] | 0.835 [0.7681, 0.9024] |
| | $C$ component | 0.658 [0.5951, 0.7206] | 0.733 [0.7047, 0.7620] | 0.835 [0.7681, 0.9024] |
| | $\tilde{S}$ component | 0.824 [0.6491, 0.9985] | 0.871 [0.7671, 0.9741] | 0.862 [0.7138, 1.0000] |
| | CAKE[PR] | 0.855 [0.7552, 0.9541] | 0.865 [0.8118, 0.9190] | **0.923** [0.8345, 1.0000] |
| | CAKE[HM] | **0.874** [0.7222, 1.0000] | **0.900** [0.8097, 0.9908] | 0.909 [0.7849, 1.0000] |

| | | ARI | AMI | ACC |
|---|---|---|---|---|
| BC | Full | 0.663 [0.6558, 0.6698] | 0.543 [0.5343, 0.5518] | 0.908 [0.9058, 0.9101] |
| | Random | 0.660 [0.6598, 0.6598] | 0.534 [0.5337, 0.5337] | 0.907 [0.9070, 0.9070] |
| | Consensus | **0.779** [0.7788, 0.7789] | **0.679** [0.6773, 0.6815] | 0.942 [0.9422, 0.9422] |
| | $C$ component | **0.779** [0.7788, 0.7789] | **0.679** [0.6773, 0.6815] | 0.942 [0.9422, 0.9422] |
| | $\tilde{S}$ component | 0.754 [0.7543, 0.7543] | 0.652 [0.6516, 0.6516] | **0.950** [0.9497, 0.9497] |
| | CAKE$^{(PR)}$ | 0.754 [0.7543, 0.7543] | 0.652 [0.6516, 0.6516] | **0.950** [0.9497, 0.9497] |
| | CAKE$^{(HM)}$ | 0.754 [0.7543, 0.7543] | 0.652 [0.6516, 0.6516] | **0.950** [0.9497, 0.9497] |
| DG | Full | 0.630 [0.6032, 0.6568] | 0.731 [0.7204, 0.7421] | 0.747 [0.7153, 0.7785] |
| | Random | 0.594 [0.5726, 0.6145] | 0.712 [0.7009, 0.7240] | 0.713 [0.6918, 0.7351] |
| | Consensus | 0.777 [0.7201, 0.8344] | 0.835 [0.8094, 0.8614] | 0.799 [0.7503, 0.8484] |
| | $C$ component | 0.774 [0.7250, 0.8239] | 0.834 [0.8111, 0.8569] | 0.784 [0.7296, 0.8376] |
| | $\tilde{S}$ component | 0.763 [0.7175, 0.8092] | 0.852 [0.8316, 0.8725] | 0.779 [0.7312, 0.8268] |
| | CAKE$^{(PR)}$ | 0.741 [0.6807, 0.8013] | 0.842 [0.8126, 0.8711] | 0.744 [0.6829, 0.8055] |
| | CAKE$^{(HM)}$ | **0.793** [0.7466, 0.8392] | **0.866** [0.8452, 0.8875] | **0.815** [0.7730, 0.8575] |
| NG | Full | 0.352 [0.3421, 0.3610] | 0.509 [0.5000, 0.5179] | 0.511 [0.4957, 0.5259] |
| | Random | 0.359 [0.3505, 0.3675] | 0.516 [0.5080, 0.5232] | 0.515 [0.4975, 0.5332] |
| | Consensus | 0.462 [0.4500, 0.4735] | 0.586 [0.5819, 0.5895] | 0.575 [0.5581, 0.5922] |
| | $C$ component | 0.473 [0.4607, 0.4863] | 0.590 [0.5831, 0.5977] | 0.578 [0.5586, 0.5973] |
| | $\tilde{S}$ component | 0.467 [0.4437, 0.4903] | 0.609 [0.5998, 0.6183] | 0.561 [0.5322, 0.5892] |
| | CAKE$^{(PR)}$ | **0.494** [0.4766, 0.5123] | **0.618** [0.6118, 0.6234] | **0.603** [0.5820, 0.6246] |
| | CAKE$^{(HM)}$ | 0.467 [0.4437, 0.4903] | 0.609 [0.5998, 0.6183] | 0.561 [0.5322, 0.5892] |
| FM | Full | 0.353 [0.3416, 0.3641] | 0.506 [0.4966, 0.5163] | 0.514 [0.4880, 0.5391] |
| | Random | 0.362 [0.3495, 0.3746] | 0.509 [0.4994, 0.5193] | 0.537 [0.5127, 0.5611] |
| | Consensus | 0.433 [0.4199, 0.4458] | 0.560 [0.5533, 0.5666] | 0.596 [0.5825, 0.6097] |
| | $C$ component | 0.427 [0.4111, 0.4421] | 0.560 [0.5517, 0.5686] | 0.596 [0.5742, 0.6177] |
| | $\tilde{S}$ component | 0.447 [0.4295, 0.4637] | 0.585 [0.5740, 0.5961] | 0.568 [0.5357, 0.5999] |
| | CAKE$^{(PR)}$ | 0.445 [0.4149, 0.4744] | 0.588 [0.5748, 0.6012] | 0.566 [0.5274, 0.6046] |
| | CAKE$^{(HM)}$ | **0.474** [0.4467, 0.5007] | **0.595** [0.5791, 0.6111] | **0.599** [0.5586, 0.6402] |
| PD | Full | 0.541 [0.5121, 0.5690] | 0.676 [0.6667, 0.6846] | 0.669 [0.6326, 0.7057] |
| | Random | 0.566 [0.5336, 0.5976] | 0.682 [0.6725, 0.6917] | 0.712 [0.6633, 0.7611] |
| | Consensus | 0.653 [0.6190, 0.6874] | 0.734 [0.7210, 0.7467] | 0.718 [0.6904, 0.7451] |
| | $C$ component | 0.626 [0.5952, 0.6563] | 0.721 [0.7078, 0.7345] | 0.704 [0.6768, 0.7316] |
| | $\tilde{S}$ component | 0.754 [0.6854, 0.8226] | **0.842** [0.8137, 0.8695] | 0.803 [0.7354, 0.8708] |
| | CAKE$^{(PR)}$ | 0.724 [0.6749, 0.7740] | 0.819 [0.7981, 0.8406] | 0.784 [0.7372, 0.8309] |
| | CAKE$^{(HM)}$ | **0.759** [0.7166, 0.8010] | **0.842** [0.8242, 0.8597] | **0.810** [0.7665, 0.8536] |
| LT | Full | 0.152 [0.1456, 0.1577] | 0.368 [0.3634, 0.3727] | 0.281 [0.2751, 0.2869] |
| | Random | 0.152 [0.1476, 0.1557] | 0.373 [0.3675, 0.3794] | 0.284 [0.2790, 0.2888] |
| | Consensus | 0.199 [0.1939, 0.2031] | 0.430 [0.4264, 0.4333] | 0.324 [0.3177, 0.3306] |
| | $C$ component | 0.210 [0.2031, 0.2169] | 0.440 [0.4348, 0.4461] | 0.333 [0.3230, 0.3434] |
| | $\tilde{S}$ component | 0.213 [0.2034, 0.2234] | 0.458 [0.4492, 0.4665] | 0.334 [0.3203, 0.3472] |
| | CAKE$^{(PR)}$ | 0.219 [0.2096, 0.2277] | 0.458 [0.4495, 0.4659] | 0.328 [0.3176, 0.3380] |
| | CAKE$^{(HM)}$ | **0.222** [0.2163, 0.2267] | **0.460** [0.4555, 0.4643] | **0.335** [0.3277, 0.3422] |
| SA | Full | 0.510 [0.4657, 0.5550] | 0.595 [0.5556, 0.6346] | 0.663 [0.6298, 0.6972] |
| | Random | 0.520 [0.4927, 0.5472] | 0.608 [0.5895, 0.6273] | 0.672 [0.6542, 0.6894] |
| | Consensus | 0.523 [0.4794, 0.5669] | 0.575 [0.5549, 0.5945] | 0.706 [0.6551, 0.7569] |
| | $C$ component | 0.523 [0.4794, 0.5669] | 0.575 [0.5549, 0.5945] | 0.706 [0.6551, 0.7569] |
| | $\tilde{S}$ component | **0.710** [0.7080, 0.7125] | **0.764** [0.7584, 0.7703] | **0.799** [0.7842, 0.8138] |
| | CAKE$^{(PR)}$ | 0.682 [0.6383, 0.7249] | 0.749 [0.7268, 0.7704] | 0.772 [0.7317, 0.8129] |
| | CAKE$^{(HM)}$ | 0.653 [0.5997, 0.7067] | 0.734 [0.7087, 0.7594] | 0.750 [0.6964, 0.8037] |

Table 1: ARI, AMI, ACC (clustering accuracy) averages and $t$-95% confidence intervals, computed over multiple independent clustering runs with varying random seeds. Results are reported for the full datasets (Full) and their filtered subsets mentioned in §5.2. Highest values in **bold**.

**Runtime.** We quantify the computational cost of CAKE and the benefit of the centroid-based Silhouette proxy by measuring runtime as a function of ensemble size $R$ and dataset size $n$ on synthetic Gaussian data (Fig. 6). The empirical trends closely match our complexity analysis (§3.4), showing approximately linear growth in $R$ and, for the centroid-based variant, in $n$.

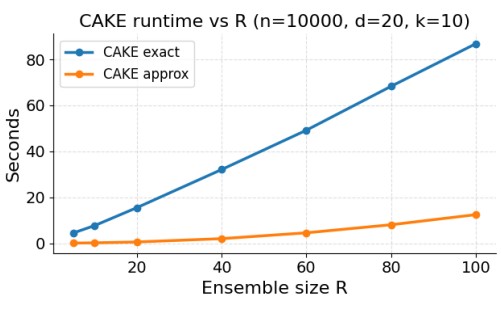 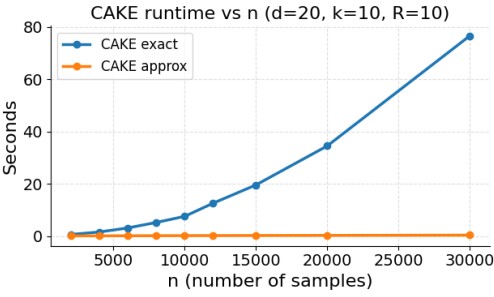

(a) Runtime vs. $R$ (ensemble size)  (b) Runtime vs. $n$ (number of samples)

Figure 6: Runtime (s) for $\text{CAKE}^{(\text{HM})}$ (Eq. 5b) using exact pairwise Silhouette scores versus the centroid-based Silhouette proxy on synthetic data. **Left** (a): runtime as a function of ensemble size $R$ ($n=10{,}000$, $d=20$, $k=10$). **Right** (b): runtime as a function of the number of samples $n$ ($d=20$, $k=10$). In both cases, the observed scaling matches the analysis in §3.4. Experiments ran on a cloud-hosted notebook with $\sim$51 GB RAM. Exact runtimes, along with correlation and mean absolute error between CAKE scores under the exact and approximate variants, are reported in Appendix C.3.

### 5.3 RESULTS

CAKE-based filtering consistently improves clustering quality. Retaining the top 70% of points by $\text{CAKE}^{(\text{HM/PR})}$ improves clustering across *all* metrics (ARI/AMI/ACC in Table 1 and Silhouette/NMI in Appendix C.5, Table 4) relative to using all points on both synthetic and real datasets, typically with tighter $t$–95% CIs. Component baselines ($\tilde{S}$ or $C$ alone) and the co-association–based consensus baseline are competitive on some datasets but are generally surpassed by CAKE, reflecting the benefit of combining stability and geometry. Accuracy rises nearly monotonically with the CAKE percentile on both synthetic and real data (Fig. 2; coverage–accuracy trade-offs between $\text{CAKE}^{(\text{PR})}$ and $\text{CAKE}^{(\text{HM})}$ on real datasets are shown in Appendix C.5, Fig. 11). Per-dataset rank correlations between CAKE and accuracy are strong (Appendix Table 4), and higher than for a bootstrap-based stability score (Appendix Fig. 13). When used for error discovery, $\text{CAKE}^{(\text{HM})}$ concentrates misclustered points more effectively than GMM posteriors $p_{\max}$ across real datasets, and a simple fusion of the two signals is competitive (Fig. 5). CAKE converges with moderate ensembles. The median per-point CAKE standard deviation decreases as $R$ grows (Fig. 3), with clear stabilization around $R \approx 30$–$40$, indicating that moderate ensembles suffice. CAKE is most informative near the appropriate number of clusters and is computationally practical. Under misspecified $k'$, ARI peaks near the ground-truth $k$, while CAKE's AURC is lowest (better ranking of correct assignments) in the same region (Fig. 4, Appendix C.5, Figs. 9–10), indicating that CAKE's ranking quality tracks clustering accuracy and is most informative near the true $k$. Runtime experiments confirm that the centroid-based Silhouette proxy closely tracks the exact computation while substantially reducing cost, and that observed scaling in $R$ and $n$ matches our complexity analysis (§3.4; Fig. 6, Appendix C.3, Table 3). Finally, additional experiments with GMM ensembles, spectral clustering, and mixed $k$-means+GMM ensembles (Appendix C.5, Fig. 12; Appendix C.4, Fig. 8) further illustrate CAKE's applicability beyond $k$-means and homogeneous ensembles.

## 6 CONCLUSION

CAKE provides an interpretable, label-free per-point confidence score by fusing cross-run assignment stability with local geometric fit. It is simple to compute from a modest ensemble of standard clustering runs and yields a scalar in $[0, 1]$ for every instance. More broadly, CAKE shows that ensemble diversity, long used for uncertainty in supervised learning, transfers to unsupervised clustering: variability across runs yields per-point confidence for robust decisions. Promising directions include calibrated confidence (e.g., with a small labeled set), integration into semi- and self-supervised pipelines (e.g., uncertainty-aware pseudo-labeling/per-sample weighting), active selection, and automatic selection of $k$ via score distributions. By making instance-level uncertainty explicit, CAKE supports safer use of clustering in labeling, anomaly detection, and decision pipelines.

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

## A    APPENDIX CAKE OUTLINE

---

**Algorithm 1: Compute CAKE Scores**

---

**Require:** Dataset $\mathcal{X}$, clustering algorithm $\mathcal{F}$, number of clustering runs $R > 1$ (partitions)
**Ensure:** Cake scores $\{CAKE[i]\}$ for all $x_i \in \mathcal{X}$
    Initialize label matrix $L \in \mathbb{N}^{n \times R}$                  ▷ Labels per point/clustering run
    Initialize Silhouette score matrix $S \in \mathbb{R}^{n \times R}$       ▷ Silhouette per point/clustering run
    Initialize Agreement Vector $A \in \mathbb{N}^n$          ▷ Pairwise (clustering) run agreements
    **for** $r = 1$ to $R$ **do**
        Run clustering $\mathcal{F}$ on $\mathcal{X}$ with random seed $r$     ▷ Ensemble diversity via seed/init/resample[7]
        Let $L[:, r] \leftarrow$ predicted labels
        Compute Silhouette scores $S[:, r]$       ▷ or centroid-based Silhouette approximations §3.2
    **end for**
    Mean Silhouette per point across the ensemble (vector) $\mu \leftarrow \frac{1}{R} \sum_{r=1}^{R} S[:, r]$
    Std Silhouette per point across the ensemble (vector) $\sigma \leftarrow \sqrt{\frac{1}{R} \sum_{r=1}^{R} (S[:, r] - \mu)^2}$
    **for** each clustering pair $(r_1, r_2)$ with $r_1 < r_2$ **do**
        Compute optimal label mapping $P^{(r_2 \to r_1)}$           ▷ Hungarian matching (pairwise)
        $A \leftarrow A + \mathbb{1}\left\{L[:, r_1] = P^{(r_2 \to r_1)}(L[:, r_2])\right\}$
    **end for**
    Compute Silhouette Component (vector) $\tilde{S} \leftarrow (\mu - \sigma)_+$        ▷ Opt. re-map with $\frac{\mu - \sigma + 1}{2}$
    Compute Stability Component (vector) $c \leftarrow \frac{2A}{R(R-1)}$
    Compute final confidence scores: $CAKE[i] \leftarrow \tilde{S}[i] \cdot c[i]$ or $\left(2\tilde{S}[i]c[i]\right) / \left(\tilde{S}[i] + c[i]\right)$
    **return** $\{CAKE[i]\}_{i=1}^{n}$

---

## B    APPENDIX THEORETICAL ANALYSIS

### B.1    MIS-RANKING BOUND

Fix two points $x_i, x_j \in \mathcal{X}$ such that $\theta_i \geq \theta_j + \gamma$ for some margin $\gamma > 0$. Recall that the empirical score $c_i$ (Eq. 4) is an order-2 U-statistic whose (Bernoulli) kernel $A_i^{(r_1, r_2)} \in [0, 1]$ (Eq. 3) has mean $\mathbb{E}\left[A_i^{(r_1, r_2)}\right] = \theta_i$ (§4.1). For each point we define the estimation errors: $e_i = c_i - \theta_i, \; e_j = c_j - \theta_j$. Using the Hoeffding's inequality for order-2 U-statistics with a kernel in $[0, 1]$ for $e_i, e_j$:

$$\Pr(e_i \geq \epsilon) \leq \exp\left(-\frac{R\epsilon^2}{2}\right), \quad \Pr(e_i \leq -\epsilon) \leq \exp\left(-\frac{R\epsilon^2}{2}\right) \quad \forall \epsilon > 0 \text{ (same for } e_j\text{)}.$$

We choose $\epsilon = \frac{\gamma}{2}$ and define the events: $E_1 = \left\{e_i \leq -\frac{\gamma}{2}\right\}, \; E_2 = \left\{e_j \geq \frac{\gamma}{2}\right\}$, with probabilities:

$$\Pr(E_1), \Pr(E_2), \text{ which are both bounded above by } \exp\left\{-R\left(\frac{(\gamma/2)^2}{2}\right)\right\} = \exp\left\{-R\frac{\gamma^2}{8}\right\}.$$

If we assume, for contradiction, that the event $\{c_i < c_j\}$ occurs while neither of $E_1$ nor $E_2$ takes place.
Then $e_i > -\gamma/2$ and $e_j < \gamma/2$, so that:

$$c_i = \theta_i + e_i \geq \theta_i - \frac{\gamma}{2} \geq \theta_j + \gamma - \frac{\gamma}{2} \geq \theta_j + \frac{\gamma}{2} \geq \theta_j + e_j = c_j \Rightarrow c_i \geq c_j, \text{ which contradicts } c_i < c_j.$$

Therefore, $\{c_i < c_j\} \subseteq E_1 \cup E_2$. Using the bounds on $E_1$ and $E_2$ obtained above:

$$\Pr[c_i < c_j] \leq \Pr[E_1] + \Pr[E_2] \leq 2\exp\left\{-R \cdot \frac{\gamma^2}{8}\right\}.$$

Hence, the probability of a mis-ranking satisfies: $\theta_i - \theta_j \geq \gamma \; \Rightarrow \; \Pr[c_i < c_j] \leq 2\exp\left\{-R\frac{\gamma^2}{8}\right\}$.

---

[7]Ensemble diversity can also be achieved by combining different algorithms that produce hard assignments.

## B.2 FALSE-STABILITY BOUND

Consider a uniform-noise observation $x_i \in \mathcal{X}$, then for each $L^{(r)} \in \mathcal{L}$, $r \in \{1, \dots, R\}$, its label $L_i^{(r)} \in L^{(r)}$ is drawn independently and uniformly from $\{1, \dots, k\}$. For such a point the Bernoulli kernel in Eq. 3 has expectation:

$$\mathbb{E}\left[A_i^{(r_1, r_2)}\right] = \theta = \frac{1}{k}.$$

Using again the Hoeffding's inequality for $c_i$ we get: $\Pr\left[c_i - \mathbb{E}\left[A_i^{(r_1,r_2)}\right] \geq \epsilon\right] \leq \exp\left\{-R\frac{\epsilon^2}{2}\right\}$. We fix a stability threshold $\tau \in \left(\frac{1}{k}, 1\right)$ and select $\epsilon = (\tau - \theta)$. Then, we have:

$$\Pr\left[c_i > \tau\right] = \Pr\left[c_i - \theta > \tau - \theta\right] \leq \exp\left\{-R\frac{(\tau - \theta)^2}{2}\right\}.$$

Since, $x_i$ is a uniform-noise point: $\theta = 1/k$, and thus:

$$\Pr\left[c_i > \tau \mid x_i \text{ uniform-noise}\right] \leq \exp\left\{-\frac{R}{2}\left(\tau - \frac{1}{k}\right)^2\right\}.$$

Additionally, if a fraction $\phi \in (0, 1)$ of the $n$ points of the dataset are uniform-noise, linearity of expectation turns the single-point bound into a dataset bound immediately:

$$\mathbb{E}\left[\#\left\{x_i \text{ uniform-noise}, c_i > \tau\right\}\right] \leq n \cdot \phi \cdot \exp\left\{-\frac{R}{2}\left(\tau - \frac{1}{k}\right)^2\right\}.$$

In more general scenarios, a noisy observation may be stochastically biased, due to proximity to dominant clusters or algorithmic bias, resulting in a non-uniform label distribution across runs. In such cases, the same concentration bound applies with this generalized $\theta \geq 1/k$:

$$Pr\left[c_i > \tau\right] \leq \exp\left\{-R\frac{(\tau - \theta)^2}{2}\right\}$$

and extends the false-positive bound beyond uniform noise to any setting where labelings follow a known or estimated marginal distribution.

**Geometric component.** An analogous mis-ranking guarantee holds for the Silhouette-based geometric component. For each point $x_i$ and run $r \in \{1, \dots, R\}$, let $s_i^{(r)} \in [-1, 1]$ denote its (possibly approximated) Silhouette score in run $r$. The randomness here comes from the clustering procedure (e.g., different random initializations), so for fixed $i$ we view $s_i^{(1)}, \dots, s_i^{(R)}$ as i.i.d. bounded random variables. Let

$$\bar{s}_i := \frac{1}{R}\sum_{r=1}^{R} s_i^{(r)}, \qquad \hat{\sigma}_i^2 := \frac{1}{R}\sum_{r=1}^{R}(s_i^{(r)} - \bar{s}_i)^2,$$

and recall that the ensemble geometric component used in CAKE is

$$\tilde{S}_i := \max\left(0, \ \bar{s}_i - \hat{\sigma}_i\right)$$

(up to the affine rescaling into $[0, 1]$ described in §3.2). The map

$$(s_i^{(1)}, \dots, s_i^{(R)}) \mapsto \tilde{S}_i$$

is a bounded-differences (Lipschitz) functional of the $s_i^{(r)}$'s, so by standard concentration inequalities for bounded independent variables (e.g., McDiarmid's inequality) $\tilde{S}_i$ concentrates around its population counterpart $g_i := \mathbb{E}[\tilde{S}_i]$ at the usual $1/\sqrt{R}$ rate, with sub-Gaussian tails. Repeating the mis-ranking argument of Appendix B.1, but with $c_i$ and $\theta_i$ replaced by $\tilde{S}_i$ and $g_i$, shows that whenever the true geometric scores of two points differ by a margin $\gamma > 0$ (i.e., $g_i \geq g_j + \gamma$), the probability that CAKE mis-ranks them (i.e., $\tilde{S}_i < \tilde{S}_j$) decays exponentially in $R\gamma^2$. A false-positive bound for $\tilde{S}_i$ can also be derived under additional assumptions on the distribution of Silhouette scores for noise points, again yielding exponentially small probabilities in $R$.

# C  APPENDIX RESULTS

## C.1  REAL-WORLD DATASETS

| Dataset (abbrev.) | Modality | $n$ | $d$ | $k$ | Preprocessing / Representation |
|---|---|---:|---:|---:|---|
| Iris (IR) | Tabular | 150 | 4 | 3 | None |
| Breast Cancer (BC) | Tabular | 569 | 10 | 2 | Standardize; PCA |
| Pendigits (PD) | Tabular | 10,992 | 16 | 10 | None |
| Letter (LT) | Tabular | 20,000 | 16 | 26 | Standardize |
| Digits (DG) | Image | 1,797 | 64 | 10 | Flatten 8×8 grayscale images |
| Fashion MNIST (FM) | Image | 60,000 | 784 | 10 | Flatten 28×28 grayscale images |
| Satimage (SA) | Remote sensing | 6,435 | 30 | 6 | PCA |
| 20 Newsgroups (NG) | Text | 18,846 | 100 | 20 | all-MiniLM-L6-v2 embeddings; PCA |

Table 2: Real-world datasets. $n$: samples; $d$: feature dimensionality after preprocessing; $k$: # classes.

## C.2  SYNTHETIC DATASETS

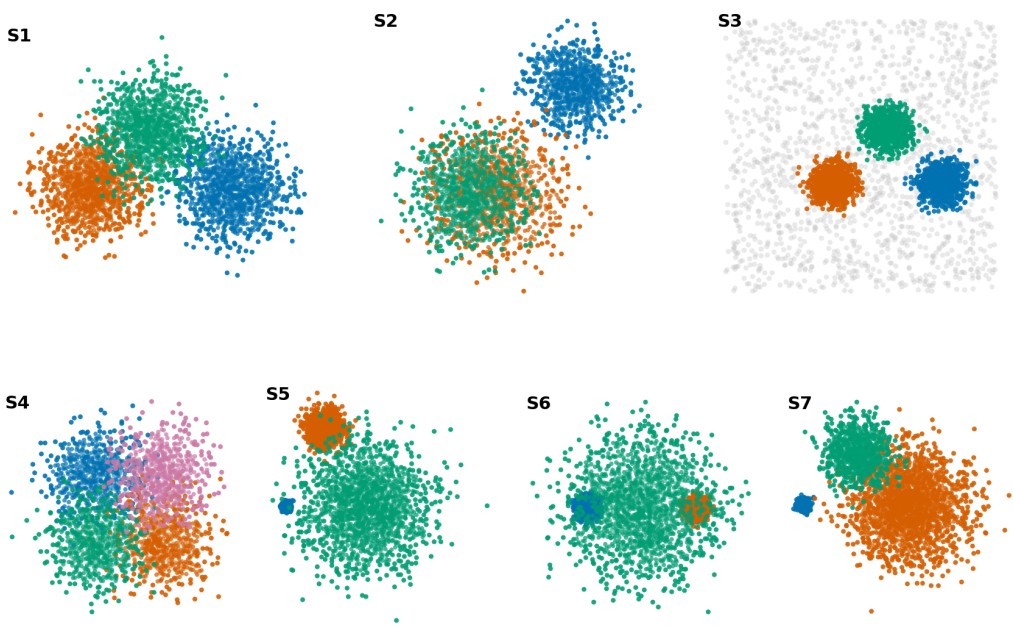

Figure 7: Synthetic datasets (S1-S7), colored by ground-truth cluster labels; noise in light gray.

## C.3  EMPIRICAL RUNTIME

| $R$ | Ensemble construction (s) | CAKE exact (s) | CAKE approx. (s) | Pearson corr. | MAE |
|---:|---:|---:|---:|---:|---:|
| 5 | 0.044 | 4.471 | 0.052 | 0.974 | 0.072 |
| 10 | 0.074 | 7.661 | 0.170 | 0.972 | 0.069 |
| 20 | 0.119 | 15.483 | 0.539 | 0.975 | 0.062 |
| 40 | 0.225 | 32.069 | 1.989 | 0.955 | 0.066 |
| 60 | 0.352 | 49.191 | 4.525 | 0.944 | 0.061 |
| 80 | 0.491 | 68.308 | 8.039 | 0.936 | 0.062 |
| 100 | 0.555 | 86.846 | 12.431 | 0.914 | 0.063 |

Table 3: Runtime (s) as a function of ensemble size $R$; Pearson correlation and Mean Absolute Error between exact and centroid-based Silhouette-proxy CAKE scores for the synthetic dataset in Fig. 6a.

## C.4 CAKE ON NON-CONVEX CLUSTER STRUCTURES

Euclidean Silhouette assumes roughly convex, isotropic clusters; on non-convex data it can mislead. We therefore use a **kernelized Silhouette** that replaces Euclidean distances by RKHS distances to cluster means under a PSD kernel $k$:

$$d_k^2(x_i, C) = k(x_i, x_i) - \frac{2}{|C|} \sum_{x_j \in C} k(x_i, x_j) + \frac{1}{|C|^2} \sum_{x_p, x_q \in C} k(x_p, x_q).$$

For each run $r$, we compute $a_i^{(r)}$ and $b_i^{(r)}$ using $d_k$, obtain per-point Silhouette scores $s_i^{(r)}$, aggregate $\mu_i = \frac{1}{R} \sum_r s_i^{(r)}$ and $\sigma_i$ across runs, and define $\tilde{S}_i = (\mu_i - \sigma_i)_+$ (we use this clipped form to preserve sparsity from negative means). Assignment-stability $c_i$ is computed by pairwise Hungarian alignment across runs (§ 3.3) and the final confidence is $\mathrm{CAKE}_i = c_i \cdot \tilde{S}_i$.

Specifically, we build an ensemble ($R{=}25$) of hard partitions using **spectral clustering** on a $k$-NN graph, jittering $n_{\text{neighbors}} \in [10, 15]$ per seed to induce diversity. We form a single *self-tuning RBF* Gram matrix $K$ with local scales $\sigma_i$ equal to the distance to the $k_{\text{nn}}{=}7$th neighbor and reuse $K$ across runs; we compute kernel Silhouette scores $s_i^{(r)}$ from $K$ and the run labels, aggregate to $(\mu_i, \sigma_i)$, and set $\tilde{S}_i = \max(\mu_i - \sigma_i, 0)$ (optionally normalizing by $\max_i \tilde{S}_i$). Finally, we compute stability $c_i$ via pairwise label agreement after Hungarian alignment and report $\mathrm{CAKE}_i^{(\mathrm{PR})} = c_i \tilde{S}_i$.

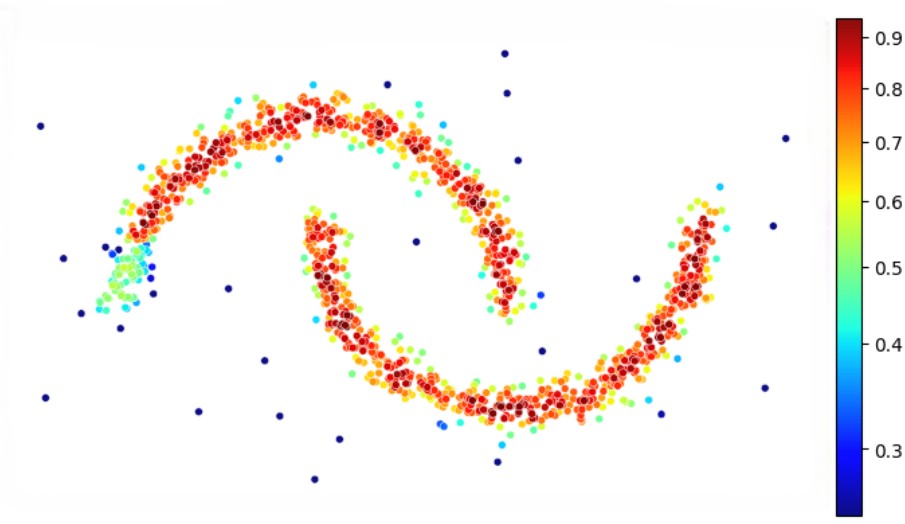

Figure 8: $\mathrm{CAKE}^{(\mathrm{PR})}$ (Eq. 5a) on non-convex structure: two moons ($n{=}1200$, noise=0.06) $+ 3\%$ uniform outliers. CAKE scores are obtained via a spectral-clustering ensemble on a $k$-NN graph ($R{=}25$ runs, $k{=}2$), jittering $n_{\text{neighbors}} \in [10, 15]$ across seeds to induce diversity. Kernelized Silhouette uses a self-tuning RBF kernel with local scale $\sigma_i$ set by the $k_{\text{nn}}{=}7$th neighbor; a single Gram matrix $K$ is built and reused across runs. Points colored by their CAKE scores (↑ **red**, ↓ **blue**).

## C.5 EXTENDED VALIDATION

Figure 9: Misspecified $k$ (synthetic datasets). Blue: ARI, red: AURC vs. $k'$. Vertical dotted line marks the ground-truth $k^*$. Ensembles use $R=30$ runs to compute CAKE; evaluation uses a separate $k$-means++ at the same $k'$. Lower AURC indicates better CAKE ranking of correct assignments.

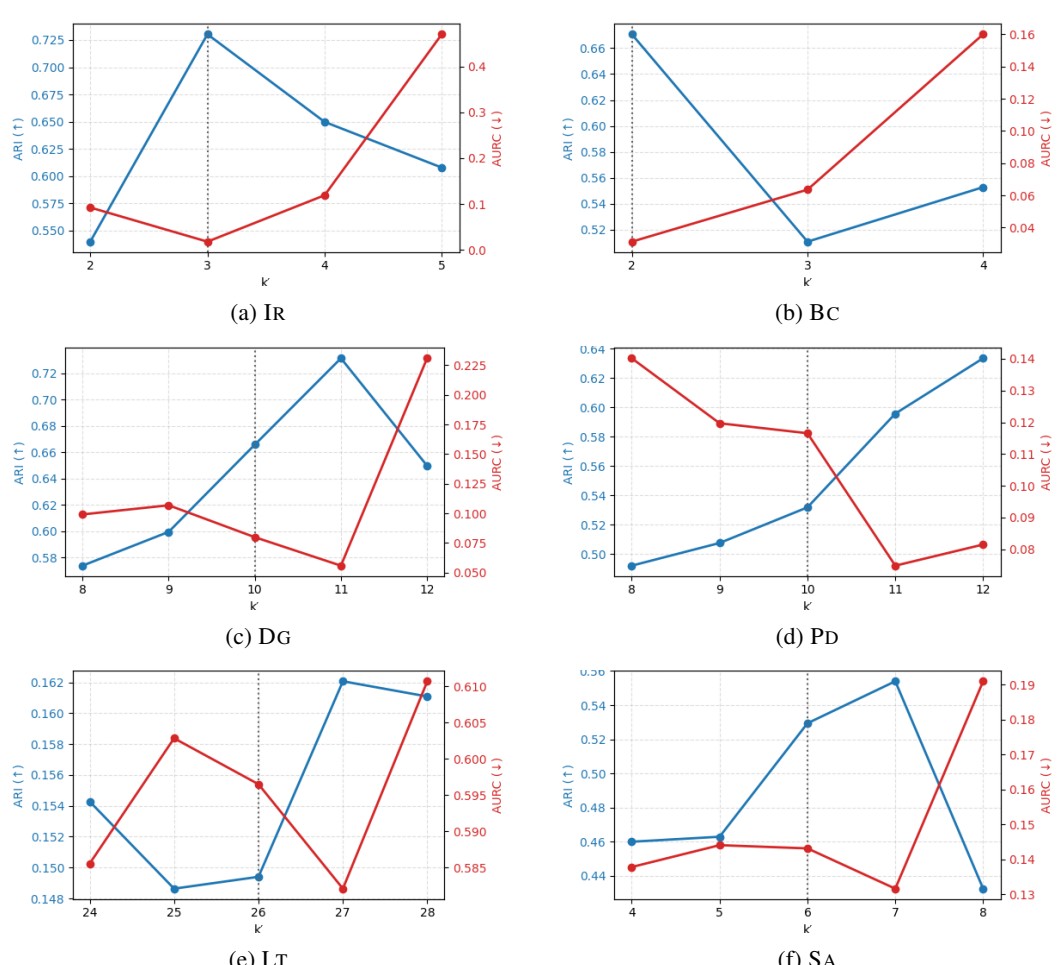

Figure 10: Misspecified $k$ (additional real datasets). Blue: ARI, red: AURC vs. $k'$. Vertical dotted line marks the ground-truth $k^*$. Ensembles use $R{=}30$ runs to compute CAKE; evaluation uses a separate $k$-means++ at the same $k'$. Lower AURC indicates better CAKE ranking of correct assignments.

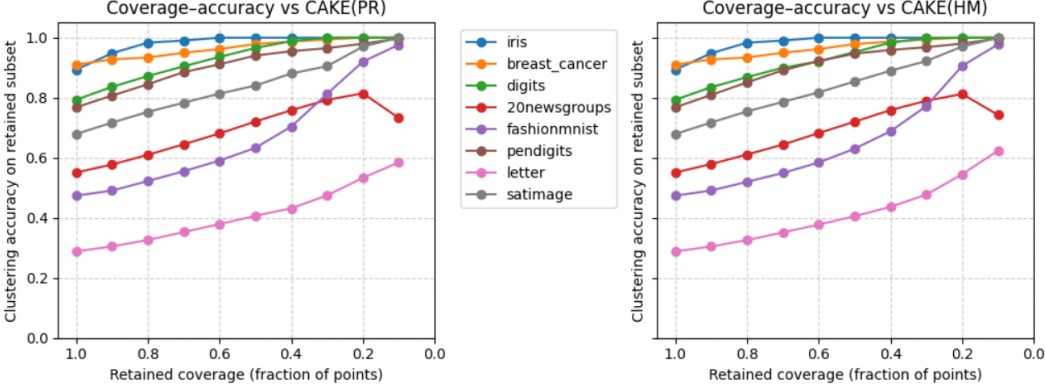

Figure 11: Coverage–accuracy trade-off comparison: clustering accuracy on the retained subset as a function of retained coverage for $\mathrm{CAKE}^{(PR)}$(Eq. 5a, left) vs. $\mathrm{CAKE}^{(HM)}$(Eq. 5b, right).

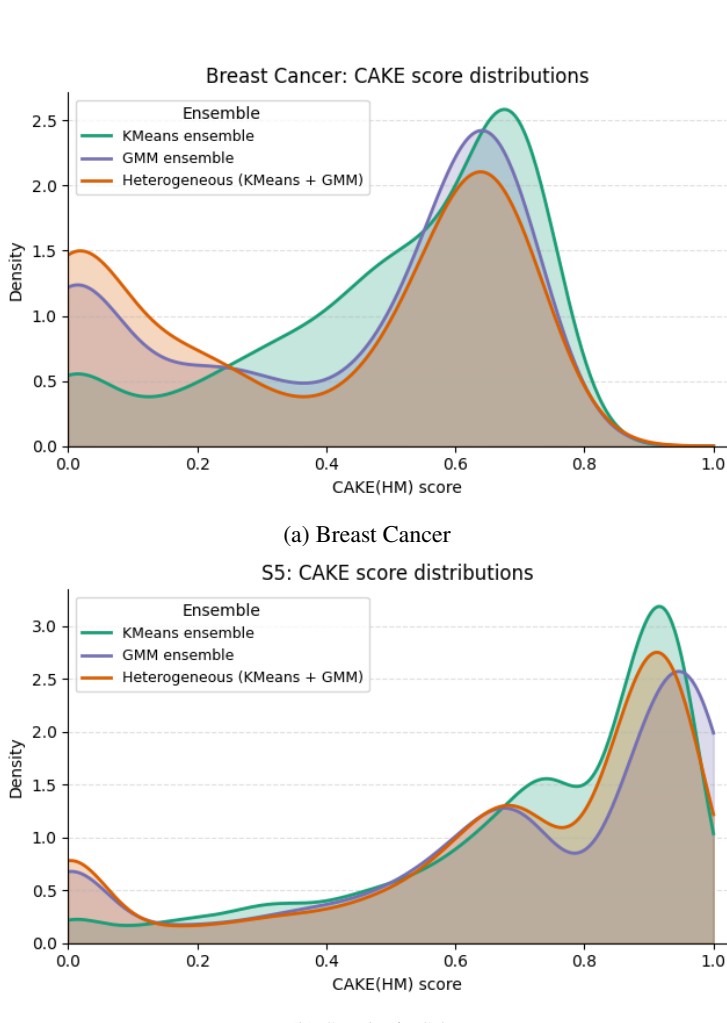

(a) Breast Cancer

(b) Synthetic S5

Figure 12: Distributions of $\text{CAKE}^{(\text{HM})}$ (Eq. 5b) confidence scores under different ensemble constructions on Bc (top) and synthetic S5 (bottom). Each plot shows kernel density estimates of per-point $\text{CAKE}^{(\text{HM})}$ scores for a homogeneous $k$-means ensemble (green), a homogeneous Gaussian Mixture (GMM) ensemble (purple), and a heterogeneous $k$-means+GMM ensemble (orange), each with $R = 10$ base clusterings. The similar, informative shapes of the three distributions across homogeneous and heterogeneous ensembles illustrate that CAKE behaves in a model-agnostic way and can be applied consistently beyond purely $k$-means-based ensembles.

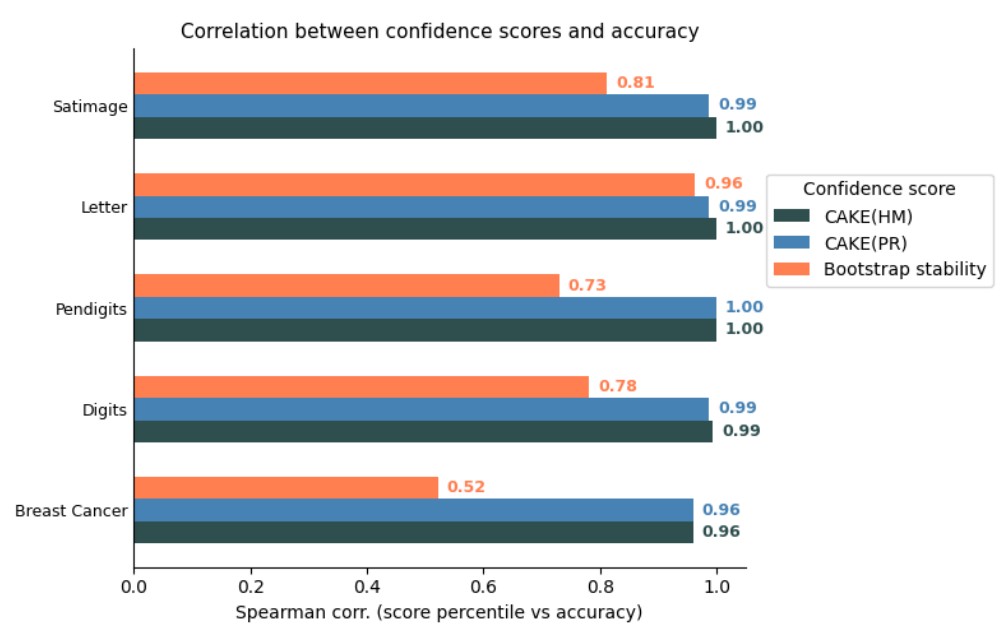

Figure 13: Illustrative comparison of the correlation between clustering accuracy and $\text{CAKE}^{\text{(PR)}}$ (Eq. 5a), $\text{CAKE}^{\text{(HM)}}$ (Eq. 5b) and bootstrap stability (same number of bootstrap replicates as the ensemble size, 80% subsampling per bootstrap replicate), mentioned in §2 (Hennig, 2007; Lange et al., 2004; Ben-Hur et al., 2002), on a subset of real datasets. $\text{CAKE}^{\text{(PR)}}$ and $\text{CAKE}^{\text{(HM)}}$ show near-perfect monotonic correlation with accuracy, clearly outperforming the bootstrap stability baseline.

| Dataset | Subset | avg Silhouette | avg NMI | CORR (CAKE $\sim$ ACC) |
|---|---|---|---|---|
| S1 | Full | 0.539 [0.5390, 0.5391] | 0.834 [0.8332, 0.8339] | |
| | Random | 0.546 [0.5460, 0.5460] | 0.834 [0.8340, 0.8340] | |
| | Consensus | 0.539 [0.5389, 0.5389] | 0.839 [0.8395, 0.8395] | |
| | $C$ component | 0.539 [0.5389, 0.5389] | 0.839 [0.8395, 0.8395] | 0.881 |
| | $\tilde{S}$ component | **0.692** [0.6921, 0.6921] | **0.983** [0.9828, 0.9828] | |
| | CAKE$^{(PR)}$ | **0.692** [0.6921, 0.6921] | **0.983** [0.9828, 0.9828] | |
| | CAKE$^{(HM)}$ | **0.692** [0.6921, 0.6921] | **0.983** [0.9828, 0.9828] | |
| S2 | Full | 0.462 [0.4499, 0.4736] | 0.606 [0.6059, 0.6070] | |
| | Random | 0.466 [0.4487, 0.4828] | 0.602 [0.5927, 0.6105] | |
| | Consensus | 0.459 [0.4499, 0.4673] | 0.628 [0.6034, 0.6533] | |
| | $C$ component | 0.459 [0.4499, 0.4673] | 0.628 [0.6034, 0.6533] | 0.975 |
| | $\tilde{S}$ component | **0.582** [0.5254, 0.6394] | 0.698 [0.6792, 0.7165] | |
| | CAKE$^{(PR)}$ | **0.582** [0.5253, 0.6384] | **0.700** [0.6803, 0.7195] | |
| | CAKE$^{(HM)}$ | **0.582** [0.5252, 0.6392] | 0.699 [0.6799, 0.7185] | |
| S3 | Full | 0.537 [0.5270, 0.5472] | 0.614 [0.6101, 0.6181] | |
| | Random | 0.503 [0.4694, 0.5357] | 0.581 [0.5357, 0.6259] | |
| | Consensus | 0.613 [0.5678, 0.6578] | 0.757 [0.7304, 0.7838] | |
| | $C$ component | 0.599 [0.5657, 0.6327] | 0.721 [0.7044, 0.7382] | 0.975 |
| | $\tilde{S}$ component | **0.639** [0.5904, 0.6867] | **0.781** [0.7593, 0.8024] | |
| | CAKE$^{(PR)}$ | 0.589 [0.5355, 0.6416] | 0.751 [0.6986, 0.8033] | |
| | CAKE$^{(HM)}$ | 0.613 [0.5858, 0.6407] | 0.777 [0.7600, 0.7934] | |
| S4 | Full | 0.434 [0.4339, 0.4339] | 0.678 [0.6778, 0.6790] | |
| | Random | 0.441 [0.4410, 0.4410] | 0.693 [0.6931, 0.6938] | |
| | Consensus | 0.391 [0.3845, 0.3981] | 0.599 [0.5946, 0.6036] | |
| | $C$ component | 0.391 [0.3845, 0.3981] | 0.599 [0.5946, 0.6036] | 0.999 |
| | $\tilde{S}$ component | **0.602** [0.6021, 0.6021] | **0.867** [0.8669, 0.8669] | |
| | CAKE$^{(PR)}$ | **0.602** [0.6021, 0.6021] | **0.867** [0.8669, 0.8669] | |
| | CAKE$^{(HM)}$ | **0.602** [0.6021, 0.6021] | **0.867** [0.8669, 0.8669] | |
| S5 | Full | 0.537 [0.4467, 0.6267] | 0.660 [0.5741, 0.7463] | |
| | Random | 0.512 [0.4115, 0.6127] | 0.634 [0.5419, 0.7268] | |
| | Consensus | 0.495 [0.4943, 0.4951] | 0.570 [0.5693, 0.5714] | |
| | $C$ component | 0.582 [0.5298, 0.6349] | 0.697 [0.6401, 0.7542] | 0.996 |
| | $\tilde{S}$ component | 0.724 [0.6001, 0.8486] | 0.855 [0.7559, 0.9536] | |
| | CAKE$^{(PR)}$ | **0.805** [0.8049, 0.8049] | 0.879 [0.8793, 0.8793] | |
| | CAKE$^{(HM)}$ | **0.805** [0.8051, 0.8051] | **0.898** [0.8980, 0.8980] | |
| S6 | Full | 0.511 [0.5106, 0.5116] | 0.474 [0.4722, 0.4750] | |
| | Random | 0.512 [0.5075, 0.5157] | 0.481 [0.4693, 0.4924] | |
| | Consensus | 0.669 [0.6571, 0.6807] | 0.586 [0.5783, 0.5946] | |
| | $C$ component | 0.675 [0.6745, 0.6748] | 0.590 [0.5898, 0.5910] | 0.963 |
| | $\tilde{S}$ component | 0.691 [0.6545, 0.7281] | 0.591 [0.5773, 0.6039] | |
| | CAKE$^{(PR)}$ | 0.697 [0.6491, 0.7454] | 0.584 [0.5576, 0.6098] | |
| | CAKE$^{(HM)}$ | **0.717** [0.6936, 0.7396] | **0.597** [0.5896, 0.6037] | |
| S7 | Full | 0.467 [0.4112, 0.5232] | 0.649 [0.5645, 0.7334] | |
| | Random | 0.466 [0.4091, 0.5222] | 0.636 [0.5467, 0.7255] | |
| | Consensus | 0.420 [0.4160, 0.4246] | 0.546 [0.5389, 0.5523] | |
| | $C$ component | 0.419 [0.4156, 0.4231] | 0.541 [0.5330, 0.5484] | 0.984 |
| | $\tilde{S}$ component | 0.581 [0.5068, 0.6557] | 0.762 [0.6531, 0.8714] | |
| | CAKE$^{(PR)}$ | **0.643** [0.5941, 0.6913] | **0.861** [0.7910, 0.9309] | |
| | CAKE$^{(HM)}$ | 0.562 [0.4828, 0.6410] | 0.741 [0.6242, 0.8580] | |
| I$_R$ | Full | 0.548 [0.5409, 0.5559] | 0.731 [0.6947, 0.7679] | |
| | Random | 0.527 [0.5200, 0.5335] | 0.703 [0.6592, 0.7470] | |
| | Consensus | 0.537 [0.5191, 0.5550] | 0.739 [0.7106, 0.7664] | |
| | $C$ component | 0.537 [0.5191, 0.5550] | 0.739 [0.7106, 0.7664] | 0.884 |
| | $\tilde{S}$ component | 0.637 [0.5766, 0.6971] | 0.873 [0.7716, 0.9745] | |
| | CAKE$^{(PR)}$ | **0.655** [0.6133, 0.6964] | 0.868 [0.8154, 0.9204] | |
| | CAKE$^{(HM)}$ | 0.648 [0.5942, 0.7015] | **0.902** [0.8134, 0.9909] | |

| | | | | |
|---|---|---|---|---|
| Bc | Full | 0.357 [0.3560, 0.3574] | 0.544 [0.5349, 0.5524] | |
| | Random | 0.360 [0.3599, 0.3599] | 0.535 [0.5346, 0.5346] | |
| | Consensus | 0.390 [0.3897, 0.3901] | **0.680** [0.6779, 0.6822] | |
| | $C$ component | 0.390 [0.3897, 0.3901] | **0.680** [0.6779, 0.6822] | 0.924 |
| | $\tilde{S}$ component | **0.557** [0.5568, 0.5568] | 0.653 [0.6526, 0.6526] | |
| | CAKE$^{(PR)}$ | **0.557** [0.5568, 0.5568] | 0.653 [0.6526, 0.6526] | |
| | CAKE$^{(HM)}$ | **0.557** [0.5568, 0.5568] | 0.653 [0.6526, 0.6526] | |
| Dg | Full | 0.178 [0.1711, 0.1857] | 0.734 [0.7233, 0.7447] | |
| | Random | 0.181 [0.1737, 0.1892] | 0.717 [0.7053, 0.7280] | |
| | Consensus | 0.213 [0.1953, 0.2311] | 0.838 [0.8123, 0.8635] | |
| | $C$ component | 0.208 [0.1904, 0.2254] | 0.837 [0.8140, 0.8591] | 0.996 |
| | $\tilde{S}$ component | 0.236 [0.2156, 0.2557] | 0.854 [0.8341, 0.8744] | |
| | CAKE$^{(PR)}$ | 0.230 [0.2110, 0.2482] | 0.844 [0.8155, 0.8731] | |
| | CAKE$^{(HM)}$ | **0.237** [0.2181, 0.2558] | **0.868** [0.8475, 0.8891] | |
| Ng | Full | 0.076 [0.0699, 0.0812] | 0.511 [0.5016, 0.5194] | |
| | Random | 0.082 [0.0800, 0.0834] | 0.518 [0.5103, 0.5254] | |
| | Consensus | 0.105 [0.1021, 0.1072] | 0.588 [0.5838, 0.5914] | |
| | $C$ component | 0.106 [0.1028, 0.1094] | 0.592 [0.5850, 0.5996] | 0.960 |
| | $\tilde{S}$ component | **0.108** [0.1047, 0.1118] | 0.611 [0.6017, 0.6201] | |
| | CAKE$^{(PR)}$ | **0.108** [0.0992, 0.1167] | **0.619** [0.6136, 0.6252] | |
| | CAKE$^{(HM)}$ | **0.108** [0.1047, 0.1118] | 0.611 [0.6017, 0.6201] | |
| Fm | Full | 0.142 [0.1373, 0.1476] | 0.507 [0.4967, 0.5164] | |
| | Random | 0.140 [0.1341, 0.1458] | 0.510 [0.4996, 0.5195] | |
| | Consensus | 0.173 [0.1662, 0.1796] | 0.560 [0.5535, 0.5668] | |
| | $C$ component | 0.173 [0.1677, 0.1786] | 0.560 [0.5519, 0.5688] | 0.890 |
| | $\tilde{S}$ component | 0.184 [0.1716, 0.1961] | 0.585 [0.5741, 0.5963] | |
| | CAKE$^{(PR)}$ | 0.185 [0.1732, 0.1967] | 0.588 [0.5750, 0.6014] | |
| | CAKE$^{(HM)}$ | **0.192** [0.1812, 0.2023] | **0.595** [0.5793, 0.6113] | |
| Pd | Full | 0.303 [0.2904, 0.3161] | 0.676 [0.6673, 0.6851] | |
| | Random | 0.291 [0.2822, 0.3005] | 0.683 [0.6733, 0.6925] | |
| | Consensus | 0.355 [0.3425, 0.3684] | 0.734 [0.7217, 0.7473] | |
| | $C$ component | 0.358 [0.3427, 0.3728] | 0.722 [0.7085, 0.7351] | 0.999 |
| | $\tilde{S}$ component | **0.395** [0.3818, 0.4087] | **0.842** [0.8141, 0.8698] | |
| | CAKE$^{(PR)}$ | 0.379 [0.3548, 0.4028] | 0.820 [0.7985, 0.8410] | |
| | CAKE$^{(HM)}$ | 0.394 [0.3777, 0.4095] | **0.842** [0.8246, 0.8601] | |
| Lt | Full | 0.141 [0.1383, 0.1433] | 0.371 [0.3665, 0.3757] | |
| | Random | 0.142 [0.1401, 0.1444] | 0.378 [0.3719, 0.3838] | |
| | Consensus | 0.185 [0.1816, 0.1880] | 0.434 [0.4304, 0.4373] | |
| | $C$ component | 0.190 [0.1864, 0.1938] | 0.444 [0.4387, 0.4500] | 0.987 |
| | $\tilde{S}$ component | **0.193** [0.1892, 0.1976] | 0.462 [0.4531, 0.4703] | |
| | CAKE$^{(PR)}$ | 0.190 [0.1849, 0.1956] | 0.461 [0.4533, 0.4697] | |
| | CAKE$^{(HM)}$ | 0.192 [0.1880, 0.1954] | **0.464** [0.4593, 0.4681] | |
| Sa | Full | 0.349 [0.3378, 0.3605] | 0.596 [0.5561, 0.6350] | |
| | Random | 0.348 [0.3350, 0.3602] | 0.609 [0.5901, 0.6279] | |
| | Consensus | 0.336 [0.3107, 0.3617] | 0.575 [0.5556, 0.5952] | |
| | $C$ component | 0.336 [0.3107, 0.3617] | 0.575 [0.5556, 0.5952] | 0.999 |
| | $\tilde{S}$ component | **0.483** [0.4688, 0.4972] | **0.765** [0.7588, 0.7707] | |
| | CAKE$^{(PR)}$ | 0.468 [0.4393, 0.4959] | 0.749 [0.7272, 0.7708] | |
| | CAKE$^{(HM)}$ | 0.443 [0.4104, 0.4765] | 0.735 [0.7092, 0.7598] | |

Table 4: Average Silhouette and NMI (Normalized Mutual Information) with $t$-95% confidence intervals and Spearman's rank correlation of CAKE percentiles with clustering accuracy percentiles. Results are reported for the full datasets (Full) and their filtered subsets mentioned in §5.2. Highest values in **bold**.

