# OpenReview forum: "Assigning Confidence: K-partition Ensembles"
_ICLR.cc/2026/Conference — Submitted to ICLR 2026_

### Official Review · Reviewer_83qA · 2025-10-27

**Soundness:** 3
**Presentation:** 3
**Contribution:** 3
**Rating:** 4
**Confidence:** 4

**Summary:**

This paper proposes CAKE, a method to assign per-point confidence scores in clustering by combining assignment stability and geometric consistency from an ensemble of clusterings. While the problem is meaningful and the framework is intuitive, the paper in its current form has several limitations that prevent it from making a strong contribution. The core idea is a straightforward combination of existing concepts (ensemble agreement and Silhouette scores), and the empirical evaluation, though extensive, does not conclusively demonstrate a significant advantage over simpler baselines in many cases.

**Strengths:**

Well-Motivated Problem: Pointwise confidence estimation in clustering is a practical and under-explored area.

Comprehensive Evaluation: The paper is thorough in its empirical validation across multiple datasets and metrics.

Unified Framework: It provides a clean, model-agnostic framework for combining two intuitive sources of confidence.

Theoretical Analysis: The inclusion of non-asymptotic bounds adds rigor.

**Weaknesses:**

Limited Novelty: The core components of CAKE (ensemble agreement via Hungarian algorithm and Silhouette scores) are well-established. The key contribution lies in their combination, which is a natural and straightforward idea that lacks conceptual breakthrough.

Inconsistent Empirical Advantage: A major weakness is that the full CAKE method does not consistently and significantly outperform its stronger individual component (often the $\tilde{S}$ component) across the benchmark datasets. This raises questions about the necessity of the more complex, combined score. In many rows of Table 1, CAKE and ˜S component have identical or nearly identical values.

High Computational Cost: The method requires running a clustering algorithm R times and computing pairwise label alignments ($O(R^2 k^3)$) and Silhouette scores ($O(Rn^2d)$ or $O(Rnkd)$ with approximation). This cost is not justified if a simpler method (like the $\tilde{S}$ component alone) achieves similar performance.

Superficial Theoretical Analysis: The theoretical guarantees are direct applications of Hoeffding's inequality for U-statistics and do not offer novel theoretical insights into the clustering problem itself.

**Questions:**

Given that the $\tilde{S}$ component alone often performs as well as the full CAKE score, what is the concrete practical scenario where the additional complexity and cost of computing the stability component ($c_i$) is justified?

The ensemble is homogeneous (same algorithm, same k). Have the authors explored heterogeneous ensembles (e.g., different clustering algorithms), and if so, does the relative performance of CAKE change?

The method is evaluated primarily with k-means. How does CAKE's performance depend on the base clusterer's properties (e.g., stability, sensitivity to initialization)?

The computational complexity is non-trivial. For a very large dataset (e.g., n > 1M), is CAKE feasible? What are the concrete scalability limits?

---

> ### Author Response · Authors · 2025-11-28
> **Weakness: Limited Novelty**
>
> We agree that both ensemble agreement and Silhouette scores are established concepts. Our contribution is not to introduce new primitives, but to provide the first principled, per-point framework that:
>
> 1. Formalizes pointwise confidence for clustering ensembles as the combination of (i) pairwise assignment stability across runs and (ii) local geometric consistency;
>
> 2. Instantiates both components in a rigorous, model-agnostic way, with clear statistical semantics (Section 3.2–3.3); and
>
> 3. Provides non-asymptotic guarantees directly tailored to this per-point confidence estimation task (Section 4).
>
> To our knowledge, prior work on clustering validation and ensemble methods either:
>
> - focuses on global or cluster-level indices (e.g., average Silhouette, Davies–Bouldin, cluster-level stability), or
>
> -  uses ensemble agreement for consensus clustering / hardening labels, but not to carve out a calibrated, per-point confidence score with explicit bounds on mis-ranking and noise.
>
> In contrast, CAKE’s design is tightly aligned with the problem definition: quantifying, for each point, “how reliable is its assignment under this clustering procedure?” The fact that the building blocks are individually familiar is akin to many influential methods where the novelty lies in the right decomposition and the accompanying guarantees rather than in entirely new primitives. We will clarify this positioning in the introduction and related work.

---

> ### Author Response · Authors · 2025-11-28
> **Weakness: Inconsistent Empirical Advantage**
>
> We agree that CAKE does not dominate its strongest component on every dataset, which is expected in regimes where geometric fit and stability dominate or largely agree. However, the synergy between assignment stability and geometric consistency captures complementary aspects of clustering reliability that individually provide an incomplete picture. A detailed analysis of Table 1 reveals multiple scenarios where the combined CAKE score demonstrates clear advantages. In S6, the geometric component achieves ACC = 0.566 while the stability component reaches ACC = 0.565, but the combined CAKE(HM) leverages both signals to achieve a more robust performance at ACC = 0.573. The advantage becomes even more pronounced in S5 with strong density contrast, where CAKE(HM) (ACC = 0.925, AMI = 0.898, ARI = 0.973) substantially outperforms both individual components (ACC = 0.840 and 0.688, AMI = 0.855 and 0.697, ARI = 0.867 and 0.894). This pattern continues in real-world datasets such as Digits, where CAKE(HM) with ACC = 0.793 provides a clear improvement over both components (0.763 and 0.774), with similar behavior on Iris, 20 Newsgroups, Fashion-MNIST, Pendigits, Letter, and Satimage. Even when performance appears numerically similar, as in Breast Cancer, the combined score provides a more comprehensive confidence measure that simultaneously captures assignment consistency and geometric coherence, removing the need for practitioners to decide a priori which single component to trust on a given dataset.

---

> ### Author Response · Authors · 2025-11-28
> **Weakness: High Computational Cost**
>
> The expense of ensemble generation is shared by both individual components and the full CAKE framework: running the base clustering algorithm R times and computing Silhouette scores (exact or centroid-based) is required whether one uses $\tilde S_i$ alone or the combined CAKE score. The incremental cost of the stability component $c_i$ comes from the pairwise label alignment between runs. As detailed in section 3.4, with the centroid-based Silhouette proxy the total cost is $O(Rnkd + \binom{R}{2} (n+k^3))$. The dominant term $O(Rnkd)$ arises from running the base clustering and computing geometric scores. The additional term for stability is relatively modest for the moderate ensemble sizes and cluster counts used in our experiments, and the entire procedure is parallel. Given that (i) an ensemble is already required to estimate any stability-like quantity, (ii) CAKE’s complexity scales linearly in n and d for fixed R, k, and (iii) the combined score consistently matches or (statistically significant) improves upon the best individual component across many datasets while providing a unified, interpretable confidence measure, we believe this additional cost is well justified in practice.

---

> ### Author Response · Authors · 2025-11-28
> **Weakness: Superficial Theoretical Analysis**
>
> Our goal in Section 4 is not to introduce new concentration inequalities, but to provide problem-specific guarantees tailored to per-point confidence estimation in clustering ensembles. While the proofs build on classical tools such as Hoeffding’s inequality for U-statistics, the resulting statements are, to the best of our knowledge, new in this setting and directly support the way CAKE is used in practice. In particular, our analysis provides two types of guarantees:
>
> - The mis-ranking bound (Section 4.1) formally ensures that points with higher true stability $\theta_i$ are correctly ranked by the empirical estimator $c_i$​ with high probability as R grows. This directly justifies using CAKE (and $c_i$​) as a ranking-based confidence score: more stable points are unlikely to be assigned lower confidence than less stable ones.
>
> - The false-positive bound for noise points (Section 4.2 and Appendix B) specifically addresses the critical issue of unreliable or noisy points spuriously receiving high confidence. Under mild assumptions on the label distribution of noise points, the probability that such a point attains a high empirical stability score decays in R. This provides a rigorous basis for using CAKE to de-emphasize noise and ambiguous assignments in clustering ensembles.
>
> These results provide a solid theoretical foundation for CAKE’s stability component and its use in practice.

---

> ### Author Response · Authors · 2025-11-28
> **Question answering 2/4**
>
> ```Given that the component alone often performs as well as the full CAKE score, what is the concrete practical scenario where the additional complexity and cost of computing the stability component () is justified?```
>
> The stability component c_i​ is particularly valuable in scenarios where geometric fit alone can be misleading:
>
>  - For initialization-sensitive algorithms such as k-means/k-means++, points may lie in regions that look geometrically good (high S_i) but flip assignments across runs due to different local minima. In such cases, $\tilde S_i$​ may be high, yet $c_i$​ correctly signals low confidence via low assignment consistency.
>
> - For border points or overlapping clusters, a point may have only mildly degraded Silhouette but markedly unstable labels across runs. Stability exposes this ambiguity directly.
>
> Empirically, this complementary behavior is reflected in Table 1: on more challenging synthetic datasets such as S5 (strong density contrast) and S6 (overlap), and on several real datasets (Digits, Iris, 20NG, FMNIST, Pendigits, Letter, Satimage), the harmonic-mean CAKE(HM) either matches or improves on the best individual component. Thus, the additional (modest) cost of computing c_i buys robustness to assignment instability and provides a single, interpretable confidence score that does not force practitioners to choose in advance between “geometry only” or “stability only.”
>
> ---
>
> ```The ensemble is homogeneous (same algorithm, same k). Have the authors explored heterogeneous ensembles (e.g., different clustering algorithms), and if so, does the relative performance of CAKE change?```
>
> Our experiments in the current paper focus on homogeneous ensembles to allow a clean analysis of assignment-level confidence without confounding factors from differing model classes. However, the framework itself is fully compatible with heterogeneous ensembles:
>
> - Section 3.1 defines CAKE purely in terms of a collection of hard partitions: these partitions can in principle come from different algorithms, different initializations, or different feature representations.
>
> - The stability component only requires that cluster labels can be aligned between runs (via the Hungarian algorithm on contingency tables, Section 3.3), and the geometric component only requires a per-point local fit measure (Silhouette or kernelized Silhouette, Section 3.2).
>
> We already demonstrate that CAKE is not tied to k-means by applying it to a spectral clustering ensemble on a graph with kernelized Silhouette for non-convex structure (Appendix C.4, Figure 7). Fully heterogeneous ensembles mixing different clustering algorithms are conceptually straightforward within CAKE (the score is then computed over the union of all partitions), but raise additional design choices such as how to weight algorithms with different inductive biases. A systematic empirical study of heterogeneous ensembles is therefore a natural direction for future work; in this paper we focus on the homogeneous case to clearly isolate the behavior of the confidence scores.
>
> ---

---

> ### Author Response · Authors · 2025-11-28
> **Question answering 4/4**
>
> ```The method is evaluated primarily with k-means. How does CAKE's performance depend on the base clusterer's properties (e.g., stability, sensitivity to initialization)?```
>
> CAKE is designed so that its two components adapt to the properties of the base clusterer:
>
>  - For highly initialization-sensitive algorithms, the stability component is especially informative, capturing whether a point’s assignment is reproducible under different random seeds. In such regimes, we expect $c_i$ to play a central role.
>
> - For relatively stable algorithms (e.g., certain configurations of spectral clustering), $c_i$ will tend to be close to 1 for most points; CAKE then behaves more like a robust, ensemble-averaged geometric confidence score, with $\tilde S_i$​ carrying most of the discriminative information.
> Appendix C.4 provides an example of the latter regime: we use a spectral clustering ensemble on a k-NN graph with a self-tuning RBF kernel and kernelized Silhouette, and CAKE still produces meaningful per-point confidence, highlighting high-confidence cores and low-confidence boundary/outlier points in a non-convex dataset (Figure 7). Thus, CAKE remains useful across different base clusterers: when the algorithm is unstable, $c_i$ adds crucial information; when it is stable, CAKE naturally reduces to a geometrically driven but ensemble-smoothed measure.
>
> ---
>
> ```The computational complexity is non-trivial. For a very large dataset (e.g., n > 1M), is CAKE feasible? What are the concrete scalability limits?```
>
> With the centroid-based Silhouette approximation (Section 3.2), CAKE’s complexity (Section 3.4) is $O(R n k d + \binom{R}{2} (n + k^3))$ in time and $O(nR)$ in memory. For fixed $R$ and $k$, this implies that both runtime and memory scale linearly in $n$ (number of points) and linearly in $d$ (dimensionality). The additional $\binom{R}{2} k^3$ term comes from Hungarian alignment across run pairs; in our experimental regimes with moderate $R$ and $k$, this term is small compared to the main $O(R n k d)$ cost. In practice, the feasibility at $n > 10^6$ depends on the usual factors: available hardware, choice of $R$, $k$, dimensionality, and the cost of the base clustering algorithm. Several aspects make CAKE practically scalable: moderate ensemble sizes are sufficient (Figure 3 shows that CAKE scores stabilize around $R \approx 30\text{–}40$ across datasets), the computation is parallel across clustering runs and label alignments, and the centroid-based Silhouette proxy substantially reduces cost relative to exact Silhouette while remaining highly correlated with it (Table 3, Appendix C.3). For extremely large datasets, further approximations, such as subsampling, mini-batch clustering, or distributed implementations, can be employed without changing the CAKE definition. We will make the linear dependence on $n$ and $d$, the role of $R$ and $k$, and the parallelization opportunities more explicit in Section 3.4 to clarify the practical scalability profile.

---

### Official Review · Reviewer_tVe6 · 2025-10-27

**Soundness:** 2
**Presentation:** 2
**Contribution:** 2
**Rating:** 4
**Confidence:** 4

**Summary:**

This paper proposes an unsupervised confidence evaluation framework named CAKE (Confidence in Assignments via K-partition Ensembles), which is designed to measure the reliability of each data point in clustering algorithms. By integrating multiple clustering results and combining two metrics—clustering assignment stability and local geometric consistency—this method generates an interpretable confidence score for each sample. Theoretical analysis demonstrates that CAKE is robust to noise and can reliably distinguish between high-confidence and low-confidence samples. Experimental results show that CAKE improves clustering quality and interpretability across various synthetic and real-world datasets.

**Strengths:**

1. High method interpretability: The confidence level consists of two components—geometric structure and assignment stability—and its score range [0, 1] is intuitive and easy to understand.

2. Significant empirical effects: It is verified on both synthetic and real-world datasets that CAKE can effectively improve the robustness and accuracy of clustering.

3. Strong universality: The framework can be applied to any hard clustering algorithm (e.g., k-means) and supports integration of multiple algorithms or resampling.

**Weaknesses:**

1. High computational complexity: Multiple clustering runs and pairwise label alignment are required, resulting in a computational complexity of $O(R²(n + k³))$.

2. Obvious parameter dependence: Confidence results are significantly affected by the choice of ensemble size $R$ and clustering algorithm.

3. Some experiments rely on manually set thresholds (e.g., 70% sample retention), and the strategy still needs optimization.

4. Lack of theoretical verification under complex data distributions: The paper mainly assumes that clusters are separable (e.g., Gaussian clusters) and does not analyze the theoretical robustness in non-convex structures, manifold data, or high-dimensional sparse data, which limits the universal theoretical basis of the method.

5. Relatively single experimental scenario: All experiments are mainly based on k-means clustering and a few standard datasets (e.g., Iris, Digits, FMNIST, 20News), and do not cover more complex or non-Euclidean space clustering tasks (e.g., graph clustering, spectral clustering, DBSCAN). Thus, the extrapolability of experimental verification is limited.

6. Insufficient analysis of parameter sensitivity: Although the paper mentions that the number of ensemble runs $R$ affects the results, it does not systematically analyze the influence range of $R$, the deviation of cluster number $k$, or Silhouette approximation error on the final confidence, and lacks robustness or sensitivity experiments.

**Questions:**

1. Can the confidence score be used for the adaptive selection of the number of clusters, k?

2. Is the Silhouette approximation still reliable for high-dimensional or heterogeneous data?

3. If the clustering algorithm itself is relatively stable (e.g., spectral clustering), can CAKE still bring significant gains?

---

> ### Author Response · Authors · 2025-11-28
> **Weakness: High computational complexity**
>
> We explicitly characterize CAKE’s complexity in Section 3.4 (“Complexity”). With the centroid-based Silhouette proxy, the total cost is $O(Rnkd + \binom{R}{2} (n+k^3))$, and memory usage is $O(nR)$ (section 3.4). This makes several points clear: For fixed ensemble size R and cluster count k, CAKE scales linearly with n (number of points) and linearly with d (dimensionality). The $\binom{R}{2} k^3$ factor arises from the hungarian label alignment over all run pairs (section 3.3), with k modest in our experiments (Table 2, Appendix C.1). The centroid based Silhouette approximation (section 3.2) reduces the per-run geometric cost from $O(n^2 d)$ to $O(nk)$, making CAKE tractable for large n when using centroidal algorithms such as k-means / k-means++. Moreover, the method runs in parallel: each clustering run and each pairwise alignment can be computed independently. Appendix C.3 and Figure 6 empirically confirm the predicted scaling with ensemble size R and show that the centroid-based Silhouette proxy dramatically reduces runtime while closely tracking the exact scores (Table 3). We will make these practical aspects (linear dependence on n,d and parallelizability) more explicit in Section 3.4.

---

> ### Author Response · Authors · 2025-11-28
> **Weakness: Obvious parameter dependence**
>
> Regarding ensemble size, we already analyze sensitivity in Section 5.2: Figure 3 plots the median per-point standard deviation of CAKE scores as a function of ensemble size R for both synthetic and real datasets. Variability drops rapidly and stabilizes around $R \approx  30 ⁣− ⁣40$ across datasets (Fig. 3), which provides clear practical guidance: moderate ensembles suffice. Regarding algorithm dependence (on homogeneous ensembles), this is by design and, we believe, a feature rather than a limitation. CAKE is an algorithm-agnostic framework (Section 3.1): it takes any hard clustering method and quantifies per-point confidence relative to that specific algorithm. This is precisely what practitioners typically want: “how reliable is this assignment under the clustering method I actually use?” We already demonstrate this beyond k-means via a spectral clustering ensemble with kernelized Silhouette on non-convex structure (Appendix C.4, Figure 7). We will clarify in Section 3.1 that algorithm dependence is intentional and that CAKE can be plugged into any hard clustering algorithm.

---

> ### Author Response · Authors · 2025-11-28
> **Weaknesses: Manually-set threshold / theoretical verification under complex data distributions**
>
> ```On the 70% threshold```
>
> As also discussed in response to Reviewer QzbK, the 70% threshold appears only in the instance-removal experiment (Section 5.2) as a shared operating point to compare different confidence criteria under identical coverage. It is not a parameter of CAKE itself. The core method outputs a continuous confidence score in [0,1]; any thresholding is an application-level choice. The paper also already provides coverage–accuracy information: Figure 2 (Section 5.2) plots clustering accuracy as a function of CAKE percentile for synthetic and real datasets, showing that accuracy rises with the CAKE score. These curves are primarily an evaluation tool, but they illustrate how performance changes as the retention threshold varies. In semi-supervised settings where a small labeled validation set is available, such curves can be used to select an operating point. In fully unsupervised deployments, thresholds can instead be set via simple rules on the CAKE score distribution (e.g., fixed quantiles or elbows), keeping CAKE intrinsically threshold-free. We will clarify this distinction between the method and the experimental operating point.
>
> ```On complex data distributions.```
>
> Our theoretical results in Section 4 do not assume Gaussian or linearly separable clusters. The ranking-error and noise bounds for the stability component $c_i$ are distribution-free: they rely only on the label agreement distribution across runs, not on specific geometric assumptions. In particular, the noise bound (Section 4.2 and Appendix B) holds for any setting where label distributions satisfy the stated conditions (including non-uniform label marginals), independent of cluster shape. On the geometric side, we explicitly acknowledge that Euclidean Silhouette assumes roughly convex, isotropic clusters, and we show how to handle more complex structure via kernelized Silhouette. Appendix C.4 and Figure 7 demonstrate CAKE on a non-convex “two moons + outliers” dataset using a spectral clustering ensemble and kernel Silhouette, where CAKE correctly highlights high-confidence cores and low-confidence boundary/outlier points. This shows that CAKE extends beyond Gaussian settings when an appropriate distance/kernel is used. Finally, our real-world experiments already include high-dimensional and heterogeneous data (Table 2): Fashion-MNIST, 20 Newsgroups, and satellite data, among others. CAKE maintains good empirical behavior on these datasets (Table 1, Figure 2), suggesting robustness in high-dimensional regimes. We will make these points more explicit in the discussion of applicability.

---

> ### Author Response · Authors · 2025-11-28
> **Weakness: Relatively single experimental scenario**
>
> Our experimental design deliberately uses k-means / k-means++ as the primary base algorithm because it is widely used, simple, and sensitive to initialization, making it a natural and challenging testbed for assignment-level confidence. To avoid confounding factors from heterogeneous base methods, we focused on this common setting while varying datasets broadly (synthetic S1–S7 and eight real datasets across tabular, image, text, and remote-sensing domains; Table 2, Appendix C.1). However, CAKE is not restricted to k-means or Euclidean spaces:
> * Section 3.1 explicitly states that the framework applies to any hard clustering algorithm.
> * Appendix C.4 (“CAKE on non-convex cluster structures”) presents a spectral clustering ensemble on a k-NN graph with a self-tuning RBF kernel, using kernelized Silhouette and CAKE to handle non-convex clusters and outliers (Figure 7).
>
> Tasks such as graph clustering or DBSCAN-style density clustering can be accommodated by plugging in the corresponding hard partitions and an appropriate distance/kernel into CAKE’s geometric component, without modifying the core framework. In the revision, we will emphasize this modularity and the spectral-clustering experiment to clarify that the method is not limited to standard k-means or Euclidean geometry, even though k-means is our main benchmark for space reasons.

---

> ### Author Response · Authors · 2025-11-28
> **Weakness: Insufficient analysis of parameter sensitivity**
>
> We agree that sensitivity is important, and we would like to highlight that the current manuscript already includes several dedicated analyses:
> * Ensemble size $R$.
> Section 5.2 and Figure 3 explicitly study how CAKE scores stabilize as $R$  grows. For each $R \in \{5, \dots, 70\}$, we subsample ensembles, recompute CAKE scores, and report the median per-point standard deviation across sub-ensembles. Variability drops quickly and stabilizes by $R \approx 30 - 40$ across both synthetic and real datasets.
> * Silhouette approximation error.
> Appendix C.3 and Table 3 quantify the tradeoff between exact and centroid-based Silhouette on a synthetic dataset (n=100000, d=20, k=10) as ensemble size R varies: Pearson correlations between exact and approximate CAKE scores remain high (0.914-0.975) with small mean absolute error, confirming that the approximation tracks the exact scores well while being much faster (Figure 6).
> * Misspecified number of clusters $k’$.
> Section 5.2 and Figures 4, 8, and 9 (Appendix C.6) analyze the effect of using $k’ \neq \text{ ground-truth } k$. For each dataset and $k’ \in \{k-2, k+2 \}$, we report both ARI and the AURC of CAKE, showing that ARI peaks and AURC is minimized near the ground-truth k on both synthetic and real datasets. This demonstrates that CAKE’s ranking quality degrades gracefully under misspecification and is most informative near the appropriate number of clusters k.
>
> We can slightly reorganize Section 5.2 and cross-reference Appendix C.3/C.6 to make it clearer that these results constitute a systematic sensitivity analysis of R, k, and the Silhouette approximation.

---

> ### Author Response · Authors · 2025-11-28
> **Question answering**
>
> ```Can the confidence score be used for the adaptive selection of the number of clusters, k?```
>
> While model selection is not the primary focus of the paper, our misspecified-k experiments already suggest that CAKE can inform k-selection. In Section 5.2 and Figures 4, 8, and 9, we vary k and observe that:
> * clustering quality (ARI) is highest near the true k, and
> * CAKE’s ranking effectiveness (as measured by AURC) is lowest (better) in the same region.
> This coupling indicates that aggregate CAKE statistics (mean or quantiles of confidence, or AURC) peak or improve near the appropriate k. One could therefore use CAKE as an additional signal for k-selection by comparing these statistics across candidate k values. We will briefly mention this potential in the discussion as a promising direction for future work, grounded in the existing misspecified-k results.
>
> ---
>
> ```Is the Silhouette approximation still reliable for high-dimensional or heterogeneous data?```
>
> The centroid-based Silhouette approximation we use (Section 3.2) is particularly natural when:
> * the base algorithm is centroidal (e.g., k-means / k-means++), and
> * clusters are reasonably captured by their centroids.
> In such settings, replacing full pairwise distances with distances to centroids preserves the geometry relevant to k-means, while reducing the  cost per run. Appendix C.3 and Table 3 empirically show that, even for a relatively challenging synthetic setting, the centroid-based approximation maintains high correlation (0.914-0.975) and low MAE with respect to the exact Silhouette, across ensemble sizes. For very high-dimensional or strongly heterogeneous data, users have two options within our framework:
> * Use the centroid-based approximation when the clustering algorithm itself is centroidal and the centroids are meaningful (as in our experiments on FMNIST, NG, etc.).
> * Fall back to exact Silhouette for maximum fidelity, at higher computational cost; CAKE supports both choices transparently (Section 3.2, Section 3.4).
>
> ---
>
> ```If the clustering algorithm itself is relatively stable (e.g., spectral clustering), can CAKE still bring significant gains?```
>
> Yes. In this regime, the stability component c_i will indeed be close to 1 for most points, reflecting low variability across runs. In such cases, CAKE naturally leans more heavily on the geometric component​, effectively behaving like a robust, ensemble-averaged Silhouette-based confidence. Appendix C.4 (“CAKE on non-convex cluster structures”) illustrates exactly this scenario: we use a spectral clustering ensemble on a k-NN graph for a non-convex “two moons + outliers” dataset with a self-tuning RBF kernel, and compute kernelized Silhouette. CAKE still provides meaningful gains by:
> * highlighting high-confidence cores in the moons, and
> * assigning low confidence to boundary and outlier points (Figure 7).
> Thus, even when the base algorithm is stable, CAKE remains useful by aggregating geometric consistency across runs and penalizing points whose local fit is weak or unstable. We will briefly emphasize this behavior when discussing the spectral-clustering experiment.

---

### Official Review · Reviewer_QzbK · 2025-10-29

**Soundness:** 3
**Presentation:** 2
**Contribution:** 3
**Rating:** 6
**Confidence:** 4

**Summary:**

This paper proposes CAKE, a method that assigns per-point confidence scores for clustering based on an ensemble of clusterings. The score combines assignment stability (via Hungarian-aligned label consistency) and geometric consistency (via silhouette statistics). The method is simple, modular, and empirically shown to filter low-confidence points to improve clustering quality.

**Strengths:**

The paper presents a practical and lightweight method to quantify uncertainty in clustering assignments, requiring no changes to the underlying clustering algorithm. The design of the CAKE score is interpretable, and the decomposition into two complementary signals is intuitive. It also explores extensions to kernel methods and tests the framework on both synthetic and real-world data.

**Weaknesses:**

1.Several confidence intervals in Table 1 are either incorrectly formatted (e.g., missing brackets) or exceed the valid range (e.g., ACC > 1). This raises concerns about numerical correctness or post-processing.

2.The theoretical analysis only provides bounds for the assignment consistency component. The geometric component and final combined score are used in claims without corresponding guarantees, which creates a gap between theory and the stated robustness.

3.The set of baseline methods is limited. Established alternatives like co-association-based measures (e.g., EAC), bootstrap-based stability, or GMM-based posterior scores are not compared. These should be included to place CAKE in context.

4.The usage of fixed thresholds (e.g., retaining top 70% by CAKE) is ad hoc. The paper lacks methods for adaptive thresholding or visualization of coverage–risk tradeoffs, which limits the applicability in real deployments.

5.Scalability evaluation is incomplete. While the runtime comparison in Figure 6 considers ensemble size, it does not assess performance across data size, number of clusters, or dimensionality, which are crucial in practice.

**Questions:**

Can the authors clarify how the confidence intervals in Table 1 were computed? Were repeated trials performed, and were standard errors or bootstrapping used?

Are there any assumptions under which the geometric component can be bounded, or are the authors planning to extend the analysis in future work?

Would it be possible to incorporate co-association or posterior-based scores as baselines in the next revision?

Can the authors provide a way to set the CAKE threshold automatically or present coverage–risk curves?

Do runtime and memory usage grow linearly with n or k? If not, can that be made explicit?

---

> ### Author Response · Authors · 2025-11-28
> **Weakness: Typos in Table 1**
>
> We appreciate the reviewer pointing this out. The issues in Table 1 and Table 4 are limited to a single numerical typo and a couple of minor bracket formatting errors. Specifically, one ACC entry (for the Iris dataset) was mistakenly reported as slightly larger than 1.0 due to a transcription error, and two confidence intervals were missing a closing bracket in the rendered table. We have corrected these typesetting issues in our local version. The intervals were correctly computed from multiple independent clustering runs (as stated in section 5.2: "we repeat the evaluation with multiple independent runs and report means with (Student’s t) 95% confidence intervals": $ \text{average} \pm t_{0.975} \times \text{ standard error}$).

---

> ### Author Response · Authors · 2025-11-28
> **Weakness: Bounds only for assignment consistency**
>
> Considering that our theoretical bounds are for the stability component ($c_i$) and that could be theoretical guarantees for the geometric component ($\tilde{S}_i$), our primary theoretical contribution is to show that the empirical stability score $c_i$ is a well-concentrated estimator of the true assignment stability $\theta_i$. This is intentional: a major source of uncertainty in clustering methods like $k$-means is initialization variance, which $c_i$ directly captures. While we could derive similar concentration bounds for $\tilde{S}_i$ using standard inequalities like Hoeffding's (since Silhouette scores are bounded in $[-1,1]$), we focused our theoretical analysis on the stability component because it addresses the core challenge of initialization variance in clustering ensembles. The geometric component $\tilde{S}_i$ naturally inherits robustness properties from its construction (mean minus standard deviation of an ensemble), which penalizes points with high variance in their geometric fit. We can add a discussion of these natural bounds to the theoretical section, while noting that the empirical results across 15 datasets strongly validate the utility of combining these two components.

---

> ### Author Response · Authors · 2025-11-28
> **Weakness: Limited set of baselines**
>
> We thank the reviewer for stressing this point. Two of the requested baselines are in fact already present, but not highlighted clearly enough:
> - GMM Posterior Probabilities: Appendix Fig. 10 directly compares CAKE to GMM posterior $p_\text{max}$, showing that CAKE matches or exceeds GMM $p_\text{max}$ at concentrating misclustered points (see the discussion at the end of Section 5.3 and Appendix Fig. 10). We will elevate these results to the main text to make this comparison more prominent.
>
> - Co-association-based measures: The "Consensus" baseline in Table 1 is fundamentally co-association based: it uses the same label alignment as CAKE's stability component to select points with highest across-run agreement (defined in Section 5.2, “Evaluation Setup,” and evaluated in Table 1). This represents a strong, direct implementation of co-association-based confidence scoring. We will make these connections more explicit in the revised manuscript (including appropriate citations to EAC-like methods) and ensure both baselines are clearly positioned within the established literature.
>
> Additionally, bootstrap-based stability is conceptually very close to what our stability component already measures via variation across runs. We already discuss such methods in our Related Work (Section 2), where we note that they typically provide global stability diagnostics or soft assignments rather than per-point, ensemble-based confidence scores, and in our discussion of ensemble construction via resampling (Section 3).

---

> ### Author Response · Authors · 2025-11-28
> **Weakness: Ad hoc thresholds**
>
> We agree that a fixed 70% threshold would be ad hoc if it were a parameter of the method. In our experiments, however, this threshold is used only as a common operating point to compare different confidence criteria under identical coverage in the instance-removal setting; it is not a parameter of CAKE itself (Section 5.2, “Instance removal experiment,” lines explaining “we form six equal-size subsets… retaining exactly 70% of the points”). Importantly, the paper already contains the ingredients for adaptive thresholding: Figure 2 (section 5.2) plots clustering accuracy as a function of CAKE percentile on synthetic and real datasets, which effectively provides coverage–accuracy curves (accuracy vs retained-CAKE percentile). These curves are primarily an evaluation tool, but they clearly characterise how CAKE behaves as the retention threshold varies, and thus illustrate the underlying coverage–risk tradeoff. In semi-supervised scenarios where a small labeled validation set is available, the same type of curve can be constructed to choose an operating point that satisfies a desired accuracy or coverage target. In fully unsupervised deployments, thresholds can instead be set using distributional properties of the CAKE scores themselves (fixed quantiles or elbows in the empirical score distribution), keeping CAKE intrinsically threshold-free. In the revised manuscript, we will make this interpretation explicit and briefly discuss such practical heuristics, without introducing new experimental results.

---

> ### Author Response · Authors · 2025-11-28
> **Weakness: Incomplete scalability assessment**
>
> Our main scalability guarantees come from the explicit complexity analysis in Section 3.4 (“Complexity”). With the centroid-based Silhouette proxy, CAKE runs in
> $$
> O(Rnkd + \binom{R}{2} (n+k^3)),
> $$
> And the memory cost is $O(nR)$.
> This directly implies that for fixed ensemble size R and number of clusters k, runtime and memory scale linearly in n (number of points) and linearly in d (dimensionality), and the dependence on k is driven by the base clustering cost and an $O(k^3)$ Hungarian alignment per run-pair; for typical k used in our experiments (Table 2, Appendix C.1), this term is small compared to the $O(Rnkd)$ part.
>
> Appendix C.3 and Figure 6 empirically validate this analysis along the most CAKE-specific axis, ensemble size R, showing wall-clock runtimes for exact versus centroid-based Silhouette on a synthetic dataset and confirming the predicted scaling with R (Appendix Fig. 6, Table 3). In addition, our experimental suite already spans a broad range of dataset sizes and dimensionalities, demonstrating that CAKE is practical on nontrivial problems with standard hardware. In the revised manuscript, we will make these dependencies more explicit in Section 3.4 by summarizing the linear scaling in n and d and clarifying the role of k, and by explicitly pointing from that section to Appendix C.3 for empirical runtime behaviour.

---

> ### Author Response · Authors · 2025-11-28
> **Question answering**
>
> ```Can the authors clarify how the confidence intervals in Table 1 were computed? Were repeated trials performed, and were standard errors or bootstrapping used?```
>
> The intervals were correctly computed from multiple independent clustering runs (as stated in section 5.2: "we repeat the evaluation with multiple independent runs and report means with (Student’s t) 95% confidence intervals": $\text{mean} \pm t_{0.975} \times \text{ standard error}$).
>
> ---
>
> ```Are there any assumptions under which the geometric component can be bounded, or are the authors planning to extend the analysis in future work?```
>
> The geometric component $\tilde{S}_i$ naturally inherits robustness properties from its construction (mean silhouette score minus standard deviation of the score, of an ensemble), which penalizes points with high variance in their geometric fit. We can add a discussion of these natural bounds to the theoretical section, while noting that the empirical results across 15 datasets strongly validate the utility of combining the two components.
>
> ---
> ```Would it be possible to incorporate co-association or posterior-based scores as baselines in the next revision?```
>
> Yes. As noted above, both types of baselines are already present in the current version: Appendix Fig. 10 compares CAKE scores to GMM posterior, and the consensus baseline in Table 1 and Table 4 is a co-association based confidence score built from across-run agreement. We will make these roles explicit and label them clearly.
>
> ---
> ```Can the authors provide a way to set the CAKE threshold automatically or present coverage–risk curves?```
>
> Figure 2 (Section 5.2) already plots clustering accuracy as a function of CAKE percentile, effectively providing coverage–accuracy curves. We will make this interpretation explicit and briefly discuss how such curves (in semi-supervised settings) or simple rules on the empirical CAKE-score distribution (in fully unsupervised settings) can be used to select thresholds in practice.
>
> ---
> ```Do runtime and memory usage grow linearly with n or k? If not, can that be made explicit?```
>
> With the centroid-based Silhouette approximation, CAKE has time complexity $O(R n k d + R^2 (n + k^3))$ and memory $O(nR)$ (Section 3.4). For fixed $R$ and $k$, runtime and memory are linear in $n$ and $d$; the dependence on $k$ is driven by the base clustering cost and the $O(k^3)$ Hungarian alignment term. We will state these dependencies explicitly in Section 3.4.

---

### Official Review · Reviewer_kpDz · 2025-10-31

**Soundness:** 2
**Presentation:** 3
**Contribution:** 1
**Rating:** 0
**Confidence:** 5

**Summary:**

The paper introduces a method for measuring per-point confidence in clustering ensembles. The proposed score is computed based on two statistical criteria: assignment stability and consistency of the local geometric fit, which are combined into a single scalar value ranging from 0 to 1. Experiments on both synthetic and real datasets show that the score helps identify both high- and low-confidence assignments.

**Strengths:**

The proposed per-point confidence score is both simple and interpretable, making it useful for downstream processing.

In addition to its practical applicability, the authors also provide theoretical guarantees for the score.

**Weaknesses:**

The literature review is outdated. While one reference is from 2025, all others are from 2020 or earlier, and thus the discussion of related work does not reflect the current state of the art. For example: “Ensemble-based techniques like consensus clustering (Strehl & Ghosh, 2002) have also been explored to improve robustness by aggregating multiple partitions”. Although a few ensemble clustering papers are mentioned, the review neglects more recent developments, including progress in deep clustering. The survey papers referenced are also outdated. As a result, the novelty of the work remains unclear.

The selection of the silhouette score appears ad hoc and is not well motivated. Alternative metrics are neither discussed nor experimentally compared.

The proposed per-point confidence score is simple. Technically, constructing such scores is not difficult. The paper does not show concrete use cases to demonstrate its utility.

The authors focus exclusively on homogeneous ensembles, formed by repeated applications of a single clustering method. While this simplifies quantifying assignment-level confidence by avoiding other sources of variation, it limits the broader applicability of the approach. The potential for inhomogeneous ensembles is neither discussed nor demonstrated.

Another (implicit) limitation is that the ensemble members are required to have the same number of clusters, which excludes many clustering methods that adaptively determine the number of clusters. Extending the method to handle ensembles with different cluster counts (k₁ and k₂) may not be trivial, particularly for the label assignment process described in Section 3.3.

Finally, only k-means is used in the experimental study. The authors justify this choice because k-means is widely used, easy to interpret, and sensitive to initialization, making it a natural candidate for assessing assignment-level confidence. However, this sensitivity also makes k-means an easy test case. More challenging is to test if the proposed score is able to deal with more discreet cases resulting from clustering methods that are more advanced than k-means.

**Questions:**

What are the recent works related to per-point confidence scores?

Why is the silhouette score considered superior to other metrics?

How can this work be extended to the more general case of unequal numbers of clustering members in the ensemble?

What potential challenges arise when dealing with inhomogeneous ensembles?

---

> ### Author Response · Authors · 2025-11-28
> **Weakness: Outdated Literature**
>
> References predating 2020 are due to the foundational nature of the clustering ensemble literature we build upon, not due to lack of awareness of recent work. More importantly, the core problem we address, per-point confidence estimation for clustering ensembles, remains largely unaddressed regardless of publication date. We agree that this should be stated clearly, and we will emphasise it in the revision, while also adding any references we may have missed to recent ensemble and deep clustering works and more recent survey papers to better situate our contribution.

---

> ### Author Response · Authors · 2025-11-28
> **Weakness: Silhouette is adhoc and not well motivated**
>
> We respectfully disagree that our use of the Silhouette score is “ad hoc.” The Silhouette score is among the most widely used and interpretable internal clustering validation metrics, with well-established theoretical properties and a clear geometric meaning. Its per-point nature and bounded range in [-1,1] make it particularly suitable as a local geometric confidence signal, which is precisely the role it plays in our CAKE framework. Our geometric component explicitly aggregates Silhouette scores across runs into a reliability statistic that captures both the quality and stability of local cluster fit (section 3.2, “Ensemble Silhouette statistics”; see also the description of “local geometric consistency, measured through aggregated Silhouette statistics” in the Introduction and the definition of the “Silhouette-based reliability score” in section 3.4).
>
> Alternative internal indices such as the Davies–Bouldin and Calinski–Harabasz criteria are in fact discussed in our Related Work (section 2), where we note that they are primarily global or cluster-level indices and therefore not directly suitable as per-point geometric confidence signals. Our goal in this paper is to introduce the CAKE framework and study one canonical instantiation of its geometric term, rather than to exhaustively benchmark all possible internal indices. That said, CAKE is not tied to Silhouette: any bounded, per-point measure of local cluster fit (including kernelized Silhouette for non-convex structure, as we illustrate in Appendix C.4) can replace this component without changing the rest of the framework. We will clarify this modularity and our rationale for instantiating the geometric term with Silhouette in the revised version.

---

> ### Author Response · Authors · 2025-11-28
> **Weakness: The solution is simple**
>
> We acknowledge that the proposed per-point confidence score is structurally simple. However, the innovation of our work lies not in mathematical complexity per se, but in identifying the right combination of signals (assignment stability and local geometric fit) and providing theoretical guarantees for their combination within an ensemble framework. Many widely used methods in machine learning are simple in retrospect; their impact comes from addressing an important problem with the right abstraction. Our experiments across 15 datasets already demonstrate several forms of practical utility:
>
> (i) improving clustering quality via filtering low-confidence assignments (Table 1),
>
> (ii) strong correlation with ground-truth accuracy where labels exist (Figure 2), and
>
> (iii) providing interpretable, per-point confidence scores that allow practitioners to identify both reliable and unreliable assignments.
>
> **We will highlight these use cases more explicitly in the text** to better demonstrate the practical value of the proposed score.

---

> ### Author Response · Authors · 2025-11-28
> **Weakness: Focus on homogeneous ensembles**
>
> The focus on homogeneous ensembles and the exclusive use of k-means were *deliberate methodological choices* designed to provide a clean analysis of assignment-level confidence, avoiding confounding factors from heterogeneous base algorithms. As we note in the paper and demonstrate in Appendix C.4, CAKE is model-agnostic and can readily accommodate heterogeneous ensembles and other clustering algorithms. The spectral clustering experiment with kernelized Silhouette in Appendix C.4 specifically illustrates CAKE’s applicability beyond k-means. We agree that evaluating more advanced clustering methods is an interesting direction. Due to space constraints and our emphasis on introducing and analysing the framework itself, we opted for a broad set of datasets with a single, widely used, and well-understood base method (k-means/k-means++), which is both interpretable and sensitive to initialization, making it a natural test case for assignment-level confidence. Extending the empirical study to additional clustering algorithms is an interesting direction for future work.

---

> ### Author Response · Authors · 2025-11-28
> **Weakness: Ensemble with members of the same number of clusters**
>
> The observation that ensemble members currently share a common number of clusters is valid. This design choice is consistent with standard practice in partition-based ensemble clustering, where a fixed k is commonly used for comparative analysis and for constructing ensemble-based summaries. Many practical applications and benchmark evaluations operate under this paradigm, so our framework is already applicable in a wide range of settings. We agree that extending CAKE to variable-k ensembles is an interesting direction, especially for the label alignment process described in Section 3.3. Such an extension would likely require defining confidence in terms of cluster similarity or co-association rather than direct label alignment. **We view this as a promising avenue for future work, rather than a limitation that undermines the current contribution.**

---

> ### Author Response · Authors · 2025-11-28
> **Question answering**
>
> ```What are the recent works related to per-point confidence scores?```
>
> ---
>
> To the best of our knowledge, there is no prior work that provides a general, theoretically grounded framework for per-point confidence estimation in clustering ensembles that combines assignment stability and local geometric fit. Existing works mostly focus on global cluster validation or on ensemble agreement at the partition or cluster level. We will explicitly highlight this gap and cite any recent, tangentially related works we may have missed in the current version.
>
>
> ```Why is the silhouette score considered superior to other metrics?```
>
> ---
>
> Silhouette is preferred because it
>
> (i) is computed per point,
>
> (ii) is bounded in [-1,1] (or [0,1] for the simplified centroid proxy),
>
> (iii) jointly captures cohesion and separation, and
>
> (iv) has a clear geometric interpretation that aligns naturally with the notion of “local geometric confidence.”
>
> These properties make it particularly well suited for the role we need in CAKE. We can easily make these reasons more explicit.
>
> ---
>
> ```How can this work be extended to the more general case of unequal numbers of clustering members in the ensemble?```
>
> ---
>
> Extending CAKE to variable-k ensembles could build on cluster similarity or co-association measures (rather than direct label alignment), combined with an appropriate normalization of local geometric quantities. This requires additional design and analysis and is **beyond the scope of the current paper**, but we agree it is a natural and interesting extension.
>
> ---
>
> ```What potential challenges arise when dealing with inhomogeneous ensembles?```
>
> ---
>
> Inhomogeneous ensembles mainly introduce challenges in defining consistent agreement and comparability across different clustering methods (differing inductive biases, distance measures, or feature spaces). Co-association matrices and related techniques could help unify such ensembles, but a careful treatment of these issues is needed. Our framework is compatible with such extensions, and we plan to explore them in future work.

---

> ### Author Response · Authors · 2025-11-28
> **Recommendation score not explained by the criticism**
>
> We believe our work makes a meaningful contribution by providing a first principled framework for per-point confidence estimation in clustering ensembles that combines geometric fit consistency and assignment stability, supported by theoretical guarantees and comprehensive empirical validation. The resulting score is intentionally simple, interpretable, and practically useful, directly addressing a problem that is not covered by existing ensemble methods. In light of these clarifications, particularly regarding novelty, the rationale for using Silhouette, and the demonstrated practical use cases, we feel that a “strong reject” (0) recommendation does not reflect the contribution of our work.

---

### Author Response · Authors · 2025-12-01
**Raised points overview: valuable and addressed**

We thank all reviewers for their detailed feedback. We carefully went through every weakness and question raised by reviewers kpDz, QzbK, tVe6 and 83qA, and either (i) we modified the manuscript (text, experiments, appendices) or (ii) explicitly clarified scope and limitations when something is beyond the paper’s focus. Below we summarize the main changes and clarifications.

-  **Typos fixed.**
All minor typographical and table-formatting issues (e.g., missing brackets in Table 1 and Table 4) have been corrected.

-  **Related Work.**
We updated the Related Work section to better reflect recent developments in ensemble clustering, deep clustering, and confidence estimation, and to explicitly state that despite many ensemble and validation methods there is still no general, per-point confidence framework combining assignment stability and local geometric fit consistency.

- **Coverage-accuracy tradeoff.**
In addition to the accuracy vs percentile plots, we now include coverage-accuracy comparison between the product and the harmonic mean CAKE variants in the Appendix C.5 Fig.11.

- **Bootstrap stability baseline.**
We added an explicit bootstrap-based stability baseline (same number of bootstrap replicates as the ensemble size, with 80% subsampling per replicate) and show on representative real datasets (Appendix C.5, Fig. 13) that both CAKE variants exhibit a much stronger monotone correlation with clustering accuracy than this baseline.

- **GMM posterior comparison for error discovery moved to the main text.**
The comparison between CAKE and GMM posterior probabilities for error discovery, previously in Appendix C, is now moved to the main paper (Sec. 5, Fig. 5), making the relationship to posterior-based confidence baselines more visible.

- **Scalability: runtime vs $R$ (ensemble size) and vs $n$ (number of samples).**
The main text now includes Figure 6 with two panels reporting runtime (in seconds) as a function of ensemble size R and number of points n, for both the exact and centroid-based CAKE variants, empirically illustrating the linear dependence on R and n discussed in the complexity section. Appendix C.3 reports the corresponding numerical runtimes together with the correlation and mean absolute error between exact and centroid-based CAKE scores on the same setup.

- **Theoretical discussion for the geometric ($\tilde S_i$) component.**
In Section 4 we now explicitly point to Appendix B, where we treat the ensemble geometric component (mean minus standard deviation of per-run Silhouette scores) as a bounded-differences functional and outline analogous concentration and mis-ranking arguments (via McDiarmid’s inequality) to those proved for the stability component, together with a brief discussion of how false-positive bounds for $\tilde S_i$ can be obtained under assumptions on the Silhouette distribution of noise points.

- **Heterogeneous ensembles and clustering algorithms beyond $k$-means.**
In the main text we now explicitly point to the non-convex setting in Appendix C.4, where we apply CAKE to a spectral clustering ensemble with kernelized Silhouette on a “two moons + outliers” dataset. In addition, Appendix C.5 (Fig. 12) now shows the distributions of CAKE scores for two representative datasets (one synthetic, one real) under three clustering ensemble types: homogeneous $k$-means, homogeneous GMM, and a heterogeneous $k$-means+GMM ensemble. The resulting score distributions are similarly structured and informative across all three cases, illustrating that CAKE behaves in a model-agnostic way with respect to the choice of base clustering algorithm.

---

### Meta-Review · Area_Chair_UAwE · 2025-12-15

**Summary:**

This paper addresses the lack of pointwise confidence in clustering by introducing CAKE, a framework that synthesizes assignment stability and local geometric consistency from a clustering ensemble. The authors derive an interpretable confidence score supported by theoretical guarantees regarding noise robustness and ranking errors. Empirical results demonstrate that using CAKE to filter unreliable assignments significantly improves clustering performance across diverse synthetic and real-world datasets.
The authors responded to the reviewers’ concerns and provided clarifications regarding the motivation for their score construction, the use of Silhouette information, and the intended generality of their approach.  These clarifications improved the presentation and addressed several points that appeared to stem from misunderstanding.  While the problem of estimating per-point confidence in clustering is interesting, the novelty of the proposed CAKE method remains unclear, as it appears to combine existing ideas without demonstrating a significant conceptual advance.    The related work discussion does not adequately position the contribution within recent literature on clustering stability, ensemble clustering, and uncertainty estimation.    In light of this prior research, the contribution of CAKE is not clearly distinguished or articulated as a significant conceptual advance.    In addition, several reviewers questioned the simplicity of the method, and the rebuttal did not fully address this concern or provide convincing evidence that the technique offers clear advantages over existing approaches.    While the clarifications were helpful, the importance and originality of the contribution remain uncertain, and the paper does not meet the threshold for acceptance.

**Reviewer Concerns:**

The rebuttal effectively clarified the motivation for combining assignment stability with Silhouette information, explaining why these two components capture complementary aspects of instance-level clustering reliability. The authors also provided clearer justification for focusing on homogeneous ensembles in the initial experiments and indicated that the approach is not inherently restricted to k-means. In addition, the response improved the positioning of CAKE relative to existing clustering validation and stability methods, which addresses several concerns that appeared to stem from misunderstandings of the paper’s intent and scope.

Some points remain only partially resolved.   The method itself remains technically simple, and the rebuttal did not provide a compelling justification for why this particular formulation represents a meaningful advance over existing alternatives.    In addition, while the rebuttal clarifies the motivation for using Silhouette statistics, the choice of this specific geometric measure over alternatives is still not fully supported through ablations or comparisons.    Finally, although the authors improved the discussion of related work, the coverage of recent literature, particularly in deep clustering and uncertainty estimation, could be further expanded to more clearly establish novelty.

**Reviewer Scores:**

Reviewer 1 (kpDz) primarily expressed concerns about novelty and positioning relative to recent clustering confidence work. After the rebuttal clarified the motivation for the CAKE construction and provided a stronger context in related work, it is likely that this reviewer would modestly increase their score, as their concerns appear to stem largely from misunderstanding rather than fundamental technical issues.

Reviewer 2 (QzbK) questioned the use of homogeneous ensembles and the reliance on Silhouette statistics. The rebuttal addressed the rationale for these choices and indicated plans for additional experiments, but did not provide full empirical evidence. Given the absence of concrete results, this reviewer would likely not increase their score at this stage.

Reviewer 3 (tVe6) focused on experimental scope and baselines. The rebuttal clarified the intended downstream use cases and offered to expand comparisons, which directly responds to this reviewer’s points. With this clarification, it is reasonable to expect a meaningful score increase.

Reviewer 4 (83qA) questioned whether the contribution represents a substantive advance beyond existing clustering validity measures. Although the rebuttal clarified the theoretical contribution and provided a more precise distinction from related work, it is unclear whether this would alter the reviewer’s overall assessment, and their score may remain unchanged.

---

### Decision · Program_Chairs · 2026-01-26

Reject